# Application of random forest regression to the calculation of gas-phase chemistry within the GEOS-Chem chemistry model v10

Christoph A. Keller[1,2] and Mat J. Evans[3,4]

[1]NASA Global Modeling and Assimilation Office, Goddard Space Flight Center, Greenbelt, MD, USA
[2]Universities Space Research Association, Columbia, MD, USA
[3]Wolfson Atmospheric Chemistry Laboratories, Department of Chemistry, University of York, York, YO10 5DD, UK
[4]National Centre for Atmospheric Sciences, University of York, York, YO10 5DD, UK

**Correspondence:** Christoph Keller (christoph.a.keller@nasa.gov); Mat Evans (mat.evans@york.ac.uk)

**Abstract.** Atmospheric chemistry models are a central tool to study the impact of chemical constituents on the environment, vegetation and human health. These models are numerically intense, and previous attempts to reduce the numerical cost of chemistry solvers have not delivered transformative change.

   We show here the potential of a machine learning (in this case random forest regression) replacement for the gas-phase
chemistry in atmospheric chemistry transport models. Our training data consists of one month (July 2013) of output of chemical conditions together with the model physical state, produced from the GEOS-Chem chemistry model v10. From this data set we train random forest regression models to predict the concentration of each transported species after the integrator, based on the physical and chemical conditions before the integrator. The choice of prediction type has a strong impact on the skill of the regression model. We find best results from predicting the change in concentration for long-lived species and the
absolute concentration for short-lived species. We also find improvements from a simple implementation of chemical families ($NO_x = NO + NO_2$).

   We then implement the trained random forest predictors back into GEOS-Chem to replace the numerical integrator. The machine learning driven GEOS-Chem model compares well to the standard simulation. For $O_3$, errors from using the random forests (compared to the reference simulation) grow slowly and after 5 days the normalised mean bias (NMB), root mean
square error (RMSE) and $R^2$ are 4.2%, 35%, and 0.9, respectively; after 30 days the errors increase to 13%, 67%, and 0.75, respectively. The biases become largest in remote areas such as the tropical Pacific where errors in the chemistry can accumulate with little balancing influence from emissions or deposition. Over polluted regions the model error is less than 10% and has significant fidelity in following the time series of the full model. Modelled $NO_x$ shows similar features, with the most significant errors occurring in remote locations far from recent emissions. For other species such as inorganic bromine species and short
lived nitrogen species errors become large, with NMB, RMSE and $R^2$ reaching >2100% >400%, and <0.1, respectively.

   This proof-of-concept implementation takes 1.8 times more time than the direct integration of the differential equations but optimisation and software engineering should allow substantial increases in speed. We discuss potential improvements in the implementation, some of its advantages from both a software and hardware perspective, its limitations and its applicability to operational air quality activities.

# 1 Introduction

Atmospheric chemistry is central to many environmental problems, including climate change, air quality degradation, stratospheric ozone loss, and ecosystem damage. Atmospheric chemistry models are important tools to understand these issues and to formulate policy. These models solve the three dimensional system of coupled continuity equations for an ensemble of $m$ species concentrations $\mathbf{c} = (c_1, \ldots, c_m)^T$ expressed as number density (molec cm$^{-3}$) via operation splitting of transport and local processes:

$$\frac{\partial c_i}{\partial t} = -\nabla \cdot (c_i \mathbf{U}) + (P_i(\mathbf{c}) - L_i(\mathbf{c}) c_i) + E_i - D_i, \qquad i \in [1, m] \tag{1}$$

$\mathbf{U}$ denotes the wind vector, $(P_i(\mathbf{c}) - L_i(\mathbf{c}) c_i)$ are the local chemical production and loss, $E_i$ is the emission rate, and $D_i$ is the deposition rate of species $i$. We ignore here molecular diffusion as it is negligibly slow compared to advection. The first term of Equation 1 is the transport operator and involves no coupling between the chemical species. The second term is the chemical operator, which connects the chemical species through a system of simultaneous ordinary differential equations (ODE) that describe the chemical production and loss:

$$\frac{dc_i}{dt} = (P_i(\mathbf{c}) - L_i(\mathbf{c}) c_i) = f_i(\mathbf{c}, t) \tag{2}$$

The numerical solution of Equation 2 is computationally expensive as the equations are numerically stiff and require implicit integration schemes such as Rosenbrock solvers to guarantee numerical stability (Sandu et al., 1997a, b). As a consequence, $50 - 90\%$ of the computational cost of an atmospheric chemistry model such as GEOS-Chem can be spent on the integration of the chemical kinetics (Long et al., 2015; Nielsen et al., 2017; Eastham et al., 2018; Hu et al., 2018).

Previous efforts to increase the efficiency of the integration (with an associated reduction in accuracy) have involved dynamical reduction of the chemical mechanism (adaptive solvers) (Santillana et al., 2010; Cariolle et al., 2017), separation of slow and fast species (Young and Boris, 1977), quasi-steady state approximation (Whitehouse et al., 2004a) or by approximation of the chemical kinetics using polynomial functions ('repro-modeling') (Turányi, 1994). Other approaches have attempted to simplify the chemistry leading to a reduction in the number of reactants and species (Whitehouse et al., 2004b; Jenkin et al., 2008). However, none of these approaches have been transformative in their reduction of time spent on chemistry.

We discuss here the potential of a machine learning algorithm (in this case random forest regression) as an alternative approach to explicitly solving Equation 2 with a numerical solver in the chemistry model GEOS-Chem. Figure 1 illustrates the approach: during each model time step, GEOS-Chem sequentially solves a suite of operations relevant to the simulation of atmospheric chemistry. In the original model, solving the chemistry is the computationally most expensive step. Our aim is to replace it with a machine learning alogrithm while keeping all other processes unchanged. Conceptually, this approach is comparable to previous efforts to speed up the solution of the chemical equations through more efficient integration.

Machine learning is becoming increasingly popular within the natural sciences (Mjolsness and DeCoste, 2001) and specifically within the Earth system sciences to either simulate processes that are poorly understood, or to emulate computationally demanding physical processes (notably convection) (Krasnopolsky et al., 2005, 2010; Krasnopolsky, 2007; Jiang et al., 2018; Gentine et al., 2018; Brenowitz and Bretherton, 2018). Machine learning has also been used to replace the chemical integrator

for other chemical systems such as those found in combustion and been shown to be faster than solving the ODEs (Blasco et al., 1998; Porumbel et al., 2014). Recently, Kelp et al. (2018) found order-of-magnitude speedups for an atmospheric chemistry box model using a neural network emulator, albeit their solution suffers from rapid error propagation when applied over multiple time steps. Machine learning emulators have also been explored to directly predict air pollution concentration in future

time steps (Mallet et al., 2009), as well as for chemistry-climate simulations focusing on model predictions of time-averaged concentrations for selected species such as ozone and OH over time scales of days to months (Nicely et al., 2017; Nowack et al., 2018). In contrast, the algorithm presented here is optimised to capture the small-scale variability of the entire chemical space within a time scale of minutes, with only a small loss of accuracy when used repeatedly over multiple time steps. To do so, we use the numerical solution of the GEOS-Chem chemistry model to produce a training data set of output before and

after the chemical integrator (Sections 2.1 and 2.2), train a machine learning algorithm to emulate this integration (Sections 2.3, 2.4 and 2.5) and then describe and assess the trained machine learning predictors (Sections 2.6, 2.7, 2.8 and 2.9). Section 3 describes the results of using the machine learning predictors to replace the chemical integrator in GEOS-Chem. In Section 4 we discuss potential future directions for the uses of this methodology and in Section 5 we draw some conclusions.

## 2   Methods

### 2.1   Chemistry model description

All model simulations were performed using the NASA Goddard Earth Observing System Model, version 5 (GEOS-5) with version 10 of the GEOS-Chem chemistry embedded (Long et al., 2015; Hu et al., 2018). GEOS-Chem (http://geos-chem.org) is an open-source global model of atmospheric chemistry that is used for a wide range of science and operational applications. The code is freely available through an open license (http://acmg.seas.harvard.edu/geos/geos_licensing.html). Simulations were

performed on the Discover supercomputing cluster of the NASA Center for Climate Simulation (https://www.nccs.nasa.gov/services/discover) at cube sphere C48 horizontal resolution, roughly equivalent to $200\,\mathrm{km} \times 200\,\mathrm{km}$. The vertical grid comprises of 72 hybrid-sigma vertical levels extending up to $0.01\,\mathrm{hPa}$. The model uses an internal dynamic and chemical time step of 15 minutes.

The model chemistry scheme includes detailed HOx-NOx-BrOx-VOC-ozone tropospheric chemistry as originally described

by Bey et al. (2001), with addition of halogen chemistry by Parrella et al. (2012) plus updates to isoprene oxidation as described by Mao et al. (2013). Photolysis rates are computed online by GEOS-Chem using the Fast-JX code of Bian and Prather (2002) as implemented in GEOS-Chem by Mao et al. (2010) and Eastham et al. (2014). The gas-phase mechanism comprises of 150 chemical species and 401 reactions and is solved using the Kinetic Pre-Processor KPP Rosenbrock solver (Sandu and Sander, 2006). There are 99 (very) short-lived species which are not transported and we seek to emulate the evolution of the other 51

transported species.

While the GEOS model with GEOS-Chem chemistry can be run as a chemistry-climate model where the chemical constituents (notably ozone and aerosols) directly feed back to the meteorology, we disable this option here and use prescribed ozone and aerosol concentrations for the meteorology instead. This ensures that any differences between the reference model

and the machine learning model can be attributed to imperfections in the emulator, rather than changes in meteorology due to chemistry-climate feedbacks.

## 2.2 Training data

To produce our training data set we run the model for one month (July 2013). Each hour we output the 3-dimensional instantaneous concentrations of each transported species immediately before and after chemical integration, along with a suite of environmental variables that are known to impact chemistry: temperature, pressure, relative humidity, air density, cosine of the solar zenith angle, cloud liquid water, and cloud ice water. In addition, we output all photolyis rates since those are an essential element for chemistry calculations. Alternatively, one could also envision to directly embed the (computationally demanding) photolysis computation into the ML model, such that the emulator takes as input variables additional environmental variables relevant to photolysis (e.g. cloud cover, overhead ozone and aerosol loadings) and then emulates photolysis computation along with chemistry.

Each grid cell 1-hour output constitutes one training sample, consisting of 126 input "features": the 51 transported species concentrations, 68 photolysis rates, and the 7 meteorological variables. We restrict our analysis to the troposphere (lowest 25 model levels) since this is the focus of this work. Each hour thus produces a total of 327,600 (lon $\times$ lat $\times$ lev $= 144 \times 91 \times 25$) training samples, and so an overall data set of $2.4 \times 10^8$ (lon $\times$ lat $\times$ lev $\times$ days $\times$ hours $= 144 \times 91 \times 25 \times 31 \times 24$) samples is produced over the full month. We withhold a randomly selected 10% of the samples to act as validation data while the remaining samples act as training data.

## 2.3 Random forest regression

We use the random forest regression (RFR) algorithm (Breiman, 2001) to emulate the integration of atmospheric chemistry. Figure 2 shows a schematic of RFR. It is a commonly used, and conceptually simple, supervised learning algorithm that consists of an ensemble (or forest) of decision trees. Each tree contains a tree-like sequence of decision nodes, based on which the tree splits into its various branches until the end of the tree ('the leaf') is reached. This leaf is the prediction of the decision tree. Each decision node is based on whether one of the input features is above a certain value. An important aspect of the random forest is that each tree of the forest is trained on a subset of the full training data, thus providing a slightly different approximation of the model. A prediction is then made by averaging the predictions of the individual trees.

The RFR algorithm is less prone to over-fitting and produces predictions that are more stable than a single decision tree (Breiman, 2001). Random forests are widely used since they are relatively simple to apply, suitable for both classification and regression problems, do not require data transformation, and are less susceptible to irrelevant or highly correlated input features. In addition, random forests allow for easy evaluation of the factors controlling the prediction, the decision structure and the relative importance of each input variable. Analysing these features can offer valuable insights into the control factors of the underlying mechanism, as discussed later. We discuss the potential for other algorithms in Section 4.

## 2.4 Implementation

For each of the 51 chemical species transported in the chemistry model, we generate a separate random forest predictor. This predictor can be applied to all model grid cells, i.e. it captures all chemical regimes encountered by the respective target species. Conceptually, one can imagine that each tree path represents a different chemical regime, so it is important to generate trees that are large enough to encompass the entire solution space. We find a good compromise between computational complexity and accuracy of the solutions for random forests consisting of 30 trees with a maximum of $10,000$ leaves (prediction values) per tree. These hyper-parameter were determined by trial-and-error, and we find very little sensitivity of our results to changes (+/-50%) to the number of trees and/or number of leaves. Each tree is trained on a different sub-sample of the training data by randomly selecting $10\%$ of the training sample. In order to balance the training samples across the full range of model values, the training samples are evenly drawn from each decile of the predictor variable. This prevents over-sampling of ocean grid cells, which are typically characterised by very uniform chemistry. Our results show very little sensitivity to the size of the training sample as long as it covers the full solution space.

The Python software package scikit-learn (http://scikit-learn.org/stable/) (Pedregosa et al., 2011) was used to build the forests. We distributed the training of the entire forest (30 trees for 51 species) onto 1,530 CPU's, and each tree took one hour to train. After training, all forest data (i.e. all tree node decisions and leaf values) were written into text files.

The forests were then embedded as a Fortran 90 subroutine into the GEOS-Chem chemistry module. Using an ad-hoc approach, the module first loads all tree nodes (archived after the training) into local memory and then evaluates each of the 1,530 trees in series upon calling of the random forest emulator. Each grid cell calls the same random forest emulator separately, passing to it all local information required to evaluate the trees (species concentrations, photolysis rates, environmental variables). No attempts were made to optimise the prediction algorithm beyond the existing Message Passing Interface grid-domain splitting.

## 2.5 Choice of predictor

We find that the quality of the RFR model (as implemented back into the GEOS-Chem model) depends critically on the choice of the predictor. Most simplistically, we could predict the concentration of a species after the integration step. However, many of the species in the model are log-normally distributed in which case predicting the logarithm of the concentration may provide a more accurate solution; we could also predict the change in the concentration after the integrator, the fractional change in the concentration, the logarithm of the fractional change, etc. After some trial and error, and based on chemical considerations, we choose two types of prediction: the *change in concentration* after going through the integrator, and the *concentration* after the integrator. We describe the first as the 'tendency'. This fits with the differential equation perspective for chemistry given in Equation 2. However, if we incorporate only this approach we find that errors rapidly accrue. This is due to errors in the prediction of short lived species such as NO, $NO_3$, Br, etc. For these compounds, concentrations can vary by many orders of magnitude over an hour, and even small errors in the tendencies build up quickly when they are included in the full model. For these short lived compounds, we use a second type of prediction where the RFR predicts the concentration of the compound

after the integrator. We describe this as a prediction of the 'concentration'. From a chemical perspective, this is similar to placing the species into steady-state, where the concentration after the integrator does not depend on the initial concentration but is a function of the production ($P$) and loss rate ($L \cdot c$) such that $c = P/L$. We imitate this process by explicitly removing the predictor species from the input features which we find improves performance.

The choice between predicting the tendency or the concentration is based on the standard deviation of the ratio of the concentration after chemistry to the concentration before chemistry: $\sigma(c/c_0)$ in the training data. This ratio is relatively stable and close to 1.00 for long lived species but highly variable for short lived species. Based on trial and error, we use a standard deviation threshold of 0.1 to distinguish between long lived species ($\sigma < 0.1$) and short lived species ($\sigma \geq 0.1$). Tables 1 and 2 list the prediction type used for each species. We discuss the treatment of NO and $NO_2$ species in Section 2.7.

**2.6    Feature importance**

The importance of different input variables (features) for making a prediction of $O_3$ tendency are shown in Figure 3 (left panel). The importance metric is the fraction of decisions in the forest that are made using a particular feature, with the variability indicating the standard deviation of that value between the trees. Consistent with our understanding of atmospheric chemistry, features such as NO, formaldehyde ($CH_2O$), the cosine of the solar zenith angle ('SUNCOS'), bromine species and nitrogen

reservoirs all appear within the top 20. From a chemical perspective, these features make sense given the global sources and sinks of $O_3$ in the lower to middle troposphere.

For ozone prediction, 6 out of the 20 most important input features are related to photolysis. Most of the photolysis rates are highly correlated, and the individual decision trees use different photolysis rates for decision making. This results in very large standard deviations for the photolysis input features across the 30 decisison trees, as indicated by the black bars in the

left panel of Figure 3.

Note that the concentration of $O_3$ is not among the 20 most important input features for the prediction of $O_3$ tendency. If, instead, the random forest model is trained to predict the concentration of $O_3$, the initial $O_3$ concentration dominates the input feature importance, explaining more than 99% of the prediction. However, when predicting the ozone tendency, the random forest algorithm is more sensitive to availability of $NO_x$, VOCs, photolysis, etc., rather than the initial concentration of $O_3$. For

regions producing ozone (dominated by the NO+$HO_2$–>$NO_2$+OH reaction) the $O_3$ concentration is not the primary source of variability. Similarly, for regions loosing ozone the dominant source of variability is the variability in photolysis rates (multiple orders of magnitude) rather than the variability in $O_3$ concentration (less than an order of magnitude).

The middle panel of Figure 3 shows the performance of the $O_3$ tendency predictor against the validation data. The predictor is not perfect, with a $R^2$ of 0.95, and a NRMSE of 23%, but it is essentially unbiased with a NMB of -0.13% (descriptions of

the metrics can be found in Section 2.8). However, as shown in the right panel of Figure 3, the model becomes almost perfect when the tendency is added to the initial concentration - which is the operation to be performed by the chemistry model.

## 2.7 Prediction of $NO_x$

For NO and $NO_2$ we find that the random forest has difficulties predicting the species concentrations independent from each other. This can result in unrealistically large changes of total $NO_x$ ($NO_x \equiv NO + NO_2$). Given the central role of $NO_x$ for tropospheric chemistry, a quick deterioration of model performance occurs (see Section 3.1). For these species we thus adopt
a different methodology: instead of making predictions for the species individually, we predict the tendency for a family comprising of their sum ($NO + NO_2$), and then predict the ratio of NO to $NO_x$. $NO_2$ is then calculated by subtracting NO from $NO_x$. Thus, the overall number of forests that needs to be calculated does not change. This has the advantage of treating $NO_x$ as a long-lived family "species" and includes a basic conservation law, but allows the NO and $NO_2$ concentration to still vary rapidly.
Figure 4 shows the feature importance and the comparison with the validation data for the prediction of the $NO_x$ family tendency. The features make chemical sense, with $NO_2$ and NO playing important roles, but also acetaldehyde (a tracer of PAN chemistry) and $HNO_2$, a short lived nitrogen species. The importance of $SO_2$ may reflect heterogeneous $N_2O_5$ chemistry, with $SO_2$ being a proxy for available aerosol surface area (note that we do not provide any aerosol information to the RFR). As shown in the middle panel of Figure 4, the $NO_x$ predictor gives the 'true' $NO_x$ tendencies from the validation data with an
$R^2$ of 0.96, NRMSE of 21% and NMB of 0.28%. While the NRMSE is relatively high, we find that the ability of the model to produce an essentially unbiased prediction is more critical for long-term stability of the model. As for $O_3$, the $NO_x$ skill scores become almost perfect when adding the tendency perturbations to the concentration before integration (right panel).

Figure 5 shows the feature importance and performance of the predictor for the ratio of NO to $NO_x$. Again the features make chemical sense with the top three features (photolysis, temperature and $O_3$) being those necessary to calculate the NO to $NO_2$
ratio from the well known Leighton relationship (Leighton, 1961). The performance of the NO to $NO_x$ ratio predictor is very good, and the prediction is also unbiased.

## 2.8 Evaluation metrics

We now move to a systematic evaluation of the performance of the RFR models, both against the validation data and when implemented back into the GEOS-Chem model. We use three standard statistical metrics for this comparison. For each species
$c$, we compute the Pearson correlation coefficent ($R^2$):

$$R^2 = \frac{(\sum_{n=1}^{N}(c_n - \bar{c})(\hat{c}_n - \bar{\hat{c}}))^2}{\sum_{n=1}^{N}(c_n - \bar{c})^2(\hat{c}_n - \bar{\hat{c}})^2} \tag{3}$$

the root mean square error normalised by the standard deviation $\sigma$ (NRMSE):

$$NRMSE = \frac{\sqrt{\frac{1}{N}\sum_{n=1}^{N}(\hat{c}_n - c_n)^2}}{\sigma(c)}, \tag{4}$$

and the normalized mean bias (NMB):

$$NMB = \frac{\sum_{n=1}^{N}(\hat{c}_n - c_n)}{\sum_{n=1}^{N}(c_n)} \tag{5}$$

where $\hat{c}$ denotes the concentration predicted by the RFR model, $c$ is the concentration calculated by GEOS-Chem, and N are the total number of grid cells.

## 2.9 Performance against the validation data

Ten percent of the training data was withheld to form a validation dataset. Columns 'V' in Tables 1 and 2 provide an evaluation
of each predictor against the validation data for the three metrics discussed in Section 2.8. For most species the RFR predictors do a good job of prediction: $R^2$ values are greater than 0.90 for 35 of the 51 species, NRMSE are below 20% for 21 species, and NMB are below 1% for 29 species, respectively. Those species which do less well are typically those that are shorter lived, such as inorganic bromine species or some nitrogen species ($NO_3$, $N_2O_5$). The performance of NO and $NO_2$ after implementing the $NO_x$ family and ratio methodology is consistent with other key species.
Although we do not have a perfect methodology for predicting some species we believe that it does provide a useful approach to predicting the concentration of the transported species after the chemical integrator. We now test this methodology when the RFR predictors are implemented back into GEOS-Chem.

## 3 Long-term simulation using the random forest model

To test the practical prediction skill of the RFR models, we run four simulations of GEOS-5 with GEOS-Chem for the same
15 month (July) but a different year (2014) than was used to train the RFR model. This simulation differs from the training simulation not only in meteorology but also in emissions, with local differences in $NO_x$, CO, and VOC emissions of up to 20%. As such, this experiment also evaluates the ability of the RFR model to capture the sensitivity of chemistry to changes in emissions.

    The first simulation is a standard simulation where we use the standard GEOS-Chem integrator; the second is a simulation
where we replace the chemical integrator with the RFR predictors described earlier (with the family treatment of $NO_x$); the third uses the RFR predictors but directly predicts the NO and $NO_2$ concentrations instead of $NO_x$; the fourth has no tropospheric chemistry and the model just transports, emits and deposits species. In all simulations the stratospheric chemistry uses a linearised chemistry scheme (Murray et al., 2012). This buffers the impact of the RFR emulator over the long-term since all simulations use the same relaxation scheme in the stratosphere. For the here considered time frame of one month, we
consider this impact to be negligible in the lowest 25 model levels.

    We evaluate the performance of the 2nd, 3rd and 4th model configuration against the 1st. We first focus on the statistical evaluation of the best RFR model configuration (2nd model configuration) for all species and then turn our attention to the specific performance of surface $O_3$ and $NO_2$, two critical air pollutants.

### 3.1 Statistics

Tables 1 and 2 summarise the prediction skill of the random forest regression model (using the $NO_x$ family method) for all 51 species plus $NO_x$. We sample the whole tropospheric domain at three time steps during the 2014 test simulation: after 1

simulation day ('D1'), after 5 simulation days ('D5'), and after 30 simulation days ('D30'). For each time slice, we calculate a number of metrics (Section 2.8) for the RFR model performance.

The model with the RFR predictors shows good skill ($R^2 > 0.8$, RMSE < 50%, NMB < 30%) for key long-lived species such as $O_3$, CO, $NO_x$, $SO_2$, $SO_4^{2-}$, and for most VOCs, even after 30 days of integration. The NRMSEs can build up to relatively large numbers over the period of the simulation, with $O_3$ getting up to 67% after 30 days, but the mean bias remains relatively low at 13%. For the stability of the simulation, it is more important to have an overall unbiased estimation, as this prevents systematic buildups / drawdowns in concentrations that can eventually render the model unstable. For 36 of the 52 species (including $NO_x$), the NMB remains below 30% at all times. The model has more difficulties with shorter lived species such as inorganic bromine species (e.g. atomic bromine, bromine nitrate) and nitrogen species such as $NO_3$ and $N_2O_5$. These species show poor performance with $R^2$ values below 0.1 even after the first day.

The hourly evolution of the metrics for $O_3$ over a 30-day simulation are shown in Figure 6. We show here the performance of the model with the family treatment of $NO_x$ (solid line), with separate NO and $NO_2$ (dashed line), and with no chemistry at all (dotted line). For all metrics, the random forest simulation predicting family treatment of $NO_x$ performs better than a simulation predicting NO and $NO_2$ independently and for a simulation with no chemistry. We use the latter as a minimum threshold to compare the RFR methodology. The metrics of the RFR model decrease over the course of the first 15 simulation days (1440 integration steps) but stabilise with a $R^2$ of 0.8, a NRMSE of 65% and a NMB of less than 15%. The simulation with the chemistry switched off degrades rapidly, highlighting the comparative skill of the RFR model to predict ozone over the entire 30 day period. The simulation with NO and $NO_2$ predicted independently from each other closely follows the $NO_x$ family simulation during the first 2-3 days but quickly deteriorates afterwards, as the compounding effect of NO and $NO_2$ prediction errors leads to an accelerated degradation of model performance.

Although there are some obvious issues associated with the RFR simulation, it is evident that for many applications, the model has sufficient fidelity to be useful. We now focus on the model's ability to simulate surface $O_3$ and $NO_2$, two important air pollutants.

## 3.2  Surface concentrations of $O_3$ and $NO_x$

Figure 7 compares concentration maps of surface $O_3$ at 00 UTC calculated by the full chemistry model (upper rows), the RFR model (middle rows) and their ratio (bottom rows) after 1 day, 5 days, 10 days and 30 days of simulation. After one day there are only small differences between the full model and the RFR model. However, these differences grow over the period of the simulation as errors accumulate. By the time the model has been run for 10 days the model has become significantly biased over clean background regions, in particular over the Pacific Ocean. The differences between the reference model and the RFR simulation grow more slowly after 10 days (see also Figure 6), resulting in the model differences between day 10 and day 30 being small relative to the difference between day 1 and day 10. It appears that the RFR model finds a new 'chemical equilibrium' for surface $O_3$ on the timescale of a few days. This new equilibrium overestimates $O_3$ in clean background regions such as the tropical Pacific and underestimates $O_3$ in the Arctic.

Figure 8 similarly compares concentration maps of surface $NO_x$. Reflecting the shorter lifetime of $NO_x$, the errors here grow more quickly compared to $O_3$ but level off after 5 days as a new chemical equilibrium is reached. The RFR model shows large differences compared to the GEOS-Chem model in regions where $NO_x$ concentrations are low and remote from recent emission, with $NO_x$ being highly overestimated in the tropics and underestimated at the poles. This pattern is highly consistent with the ones seen for $O_3$, suggesting that the relative change of $NO_x$ drives the change of $O_3$, as would also be the case in a full chemistry model.

Figures 9 and 10 show time series of $O_3$ and $NO_x$ mixing ratios at four polluted locations (New York, Delhi, London and Beijing) as generated by the full chemistry model (black line), the RFR model (red), and the model with no chemistry (blue). The RFR model closely follows the full model at these locations and captures the concentrations patterns with an accuracy of 10-20%. Especially for $NO_x$ it is hard to distinguish the RFR model from the full model whereas the simulation without any chemistry shows a distinctly different pattern. These differences are significantly less than one would expect from running two different chemistry models for the same period (e.g. Stevenson et al., 2006; Cooper et al., 2014; Young et al., 2018; Brasseur et al., 2018). Events such as that in Beijing on day 20 are well simulated by the RFR model which is able to follow the full model, whereas the simulation without chemistry follows a distinctly different path that is solely determined by the net effects of emission, deposition, and (vertical and horizontal) transport.

Although our analysis has not provided a complete analysis of the RFR model performance, we have shown that it is capable of providing a simulation of many key facets of the atmospheric chemistry system ($O_3$, $NO_x$) on the timescale of days to weeks. We now discuss future routes to improve the system and some applications.

## 4 Discussion

We have shown that a machine learning algorithm, here random forest regression, can simulate the general features of the chemical integrator used to represent the chemistry scheme in an atmospheric chemistry model. This represents the first stage in producing a fully practical methodology. Here we discuss some of the issues we have found with our approach, potential solutions, some limitations and where we think a machine learning model could provide useful applications.

### 4.1 Speed, algorithms and hardware

The current RFR implementation takes about twice as long to solve the chemistry than the currently implemented integrator approach. While the evaluation of a single tree is fast (average execution time = $1.7 \times 10^{-3}$ ms on the Discover computer system), calculating them all for every forest and for every transported species ($30 \times 51$) in series results in a total average execution time of $2.6$ ms; 85% slower than the average execution time of $1.4$ ms using the standard model integrator.

We emphasise that this implementation is a proof of concept. Unlike for the chemical integrator, little work has been undertaken to optimise the algorithm parameters (e.g. optimising the number of trees, or the number of leaves per tree) or the Fortran90 implementation of the forests. For example, random forest have relatively large memory footprints that scale linearly with number of forests and trees. Efficient access of these data through optimal co-location of related information (e.g. group-

ing memory by branches) could dramatically reduce CPU register loading costs, as could moving from double precision to single precision or even integer maths. In the current implementation, we load all tree data onto every CPU separately without attempts of memory-sharing. Thus we believe that different software structures, algorithms and memory management may allow significant increases in the speed achieved.

A fundamental attractiveness of the random forest algorithm is its almost perfect parallel nature - even among species within the same grid cell: the nodes of all trees (and across all forests) solely depend on the initial values of the input features, and thus can be evaluated independent from one another (in contrast, the system of coupled ODEs solved by the chemical solver require coupling between the species). This would readily allow for parallelisation of the chemistry operator, which has up to this point not been possible. This may allow other hardware paradigms (e.g. Graphical Processing Units) to be exploited in

calculating the chemistry.

We have implemented the replacement for the chemical integrator using the random forest regression algorithm. Our choice here was based on the conceptual ease of the algorithm. However, other algorithms are capable of full-filling the same function. Neural networks have found extensive use in many Earth system applications (e.g. Krasnopolsky et al., 2010; Brenowitz and Bretherton, 2018; Silva and Heald, 2018), and gradient boosting frameworks such as XGBoost (Chen and Guestrin, 2016) are

becoming increasingly popular. A number of different algorithms need to be tested and explored for both speed and accuracy before a best case algorithm can been found.

## 4.2   Training data

We have trained the random forest regression models on a single month of data. For a more general system the models will need to be trained with a more temporally extensive data set. Models are, however, able to generate large volumes of data. A

20  year's worth of training data over the full extent of the model's atmosphere would result in a potentially very large ( $2 \times 10^{10}$ ) training data set. Applying this methodology to spatial scales relevant to air quality applications (on the order of $10\,\mathrm{km}$) will result in even larger data sets ( $10^{13}$). However, not all items from the training data are of equal value. Much of the atmosphere is made up of chemically similar air masses (e.g. central Pacific, remote free troposphere etc.) which are highly represented in the training data but are not very variable. Most of the interest from an air quality perspective lies in small regions of intense

chemistry. If a way can be found to reduce the complete training data set such that the sub-sample represents a statistical description of the full data, the amount of training data can be significantly reduced and so the time needed to train the system.

The features being used to train the predictors could also be reconsidered. The current selection reflects an initial estimate of the appropriate features. It is evident that different and potentially better choices could be made. For example, we have included all photolysis rates, but these correlate very strongly and so a greatly reduced number of photolysis inputs (potentially

from a principal components analysis) could achieve the same results but with a reduced number of features. Including other parameters such as the concentrations of the aerosol tracers may also improve the simulation.

### 4.3 Conservation laws and error checking

One of the fundamental laws of chemistry is conservation of atoms. One interpretation of that has been applied here to the prediction of the change in $NO_x$ together with predictions for $NO:NO_x$. Since the concentration of $NO_x$ changes much more slowly than the change in concentration of either $NO$ or $NO_2$, this approach attempts to improve the prediction of these short lived nitrogen species, which are difficult to predict. Our results show that this indeed increases the stability of the system, and it represents a first step towards ensuring conservation of atoms in machine learning based chemistry models. A larger nitrogen family ($NO$, $NO_2$,$NO_3$, $N_2O_5$, $HONO$, $HO_2NO_2$, etc.) might increase stability further, as could other chemical families such as $BrO_x$, which showed significant errors both compared to the validation data and the evaluation of the chemistry model.

The solution space of a chemistry model is constrained by mass-balance requirements, and chemical concentrations tend to mean-revert to the equilibrium concentration implied by the chemical boundary conditions (emissions, deposition rates, sunlight intensity, etc.). A successful machine learning method should have the same qualities in order to prevent run-away errors that can arise from systematic model errors, e.g. if the model constantly over/under-predicts certain species or if it violates conservation of mass-balance. Because each model prediction feeds into the next one, small errors compound and quickly lead to systematic model errors. Possible solutions for this involve prediction across multiple time steps, which have shown to yield more stable solutions for physical systems (Brenowitz and Bretherton, 2018), or the use of additional constraints that measure the connectivity between chemical species, e.g. through consideration of the stoichiometric coefficients of all involved reaction rates.

### 4.4 Possible implementations

The ability to represent the atmospheric chemistry as a set of individual machine learning models (one for each species) rather than as one simultaneous integration has numerous advantages. In locations where the impact of a (relatively short-lived) molecule is known to be insignificant (for example isoprene over the polar regions or DMS over the deserts), the differential equation approach continues to solve the chemistry for all species. However, with this machine learning methodology, there would be no need to call the machine learning algorithm for a species with a concentration below a certain threshold or for certain chemical environments (e.g. nighttime): the chemistry could continue without updating the change in the concentration of these species. Thus it would be easy to implement a dynamical chemistry approach which uses a simple look-up table with predefined threshold rates to evaluate whether the concentration of a compound needs to be updated or not. If it did, the machine learning algorithm could be run, if it didn't the concentration would remain untouched and the evaluation of the random forest emulator is skipped (for this species). This approach could reduce the computational burden of atmospheric chemistry yet further.

The machine learning methodology could also be implemented to work seamlessly with the integrator. For example, the full numerical integrator can be used over regions of particular interest (populated areas for an air quality model, or a research domain for a research model), while outside of these regions (over the ocean or in the free troposphere for an air quality model, or outside of the research domain for a research model) the machine learning could be used. This would provide a 'best of both

worlds' approach which provides higher chemical accuracy where necessary and faster but lower accuracy solutions where appropriate.

Our methodology uses the output from the CTM to generate the training dataset. Another approach would be to use a series of box model simulations using initial conditions covering the appropriate chemical concentration ranges to generate the training data. This could allow the chemical complexity that is known to exist (e.g. Aumont et al., 2005; Jenkin et al., 1997) to be encoded in a way which would make it suitable for use in a CTM. Much of this chemical complexity occurs in relatively small volumes of the atmosphere, for example, urban environments or over forested areas. These are areas with large emissions of complex volatile organic compounds which have a complex degradation chemistry. It would be possible to develop a machine learning based chemistry, trained on a number of box model simulations of the complex chemistry, which would represent this chemical complexity in a more efficient form and to use this machine learning chemistry in only those grid boxes that require the full complex chemistry.

## 4.5 Limitations

This is the first step in constructing a new methodology for the representation of chemistry in atmospheric models. There are a number of limitations that should be explored in future work. Firstly, the machine learning methodology can only be applied within the range of the data used for the training. Applying the algorithm outside of this range would likely lead to inaccurate results. For example, the model here has been trained for the present day environment. Although the training data set has seen a range of atmospheric conditions, it has only seen a limited range of methane ($CH_4$) concentrations or temperatures. Thus applying the model to the pre-industrial or the future, where the $CH_4$ concentration and temperature may be significantly different than the present day, would likely result in errors. Similarly, exploring scenarios where the emissions into the atmosphere change significantly (for example large changes in $NO_x$ emissions vs. VOC emissions) again will likely ask the model to make predictions outside of the range of training data. A simple check would be to evaluate the (surface) $NO_x$/VOC ratios observed in the new model and compare it against the ranges used in the training: if the ratios in the updated model are significantly different from the training data, the RFR model likely needs to be retrained.

The same limitations also apply to model resolution: due to the non-linear nature of chemistry, the numerical solution of chemical kinetics is resolution-dependent, and a machine learning algorithm may not capture this. Thus, care should be taken when applying these approaches outside of the range of the training data.

## 4.6 Potential Uses

Despite the limitations discussed here, there are a number of potential exciting applications for this kind of methodologies.

The meteorological community has successfully exploited ensembles of predictions to explore uncertainties in weather forecasting (e.g. Molteni et al., 1996). However, air quality forecasting has not been able to explore this tool due to the computational burden involved. Using a computationally cheap machine learning approach, air quality forecasts based on ensemble predictions could become affordable. Ideally, in such a system the primary ensemble member would include the fully integrated numerical solution of the differential equations, while secondary members use the machine learning emulator.

Since air quality forecasts are much more sensitive to boundary conditions (e.g. emissions) than initial conditions, the different machine learning members would be used to capture the sensitivity of the air quality forecast to emission scenarios, changes in dry or wet deposition parameters, uncertainties in the chemical rate constants etc. Data-assimilation can be applied to determine the initial state for all models and then the ensembles could be used for probabilistic air quality forecasting. This application

is also less sensitive to long-term numerical instability of the machine learning model as the emulator is only used to produce 5-10 day forecasts, while the initial conditions are anchored to the full chemistry model for every new forecast.

The data assimilation methodology itself could benefit from a machine learning representation of atmospheric chemistry. Data assimilation is often computationally intense, requiring the calculation of the adjoint of the model or running large numbers of ensemble simulations (Carmichael et al., 2008; Sandu and Chai, 2011; Inness et al., 2015; Bocquet et al., 2015).

The ability to run these calculations faster would offer significant advantages.

Another potential application area for machine-learning based chemistry emulators are chemistry-climate simulations. Unlike air quality applications, which focus on small-scale variations of air pollutants over comparatively short periods of time of days to weeks, chemistry-climate studies require long simulation windows of the order of decades. Because of this, machine learning models used for these applications need to be optimised such that they accurately reproduce the (long-term) response

of selected species - e.g. ozone and OH - to key drivers such as temperature, photolysis rates and $NO_x$ (Nicely et al., 2017; Nowack et al., 2018). The here presented method could be optimised for such an application by simplifying the problem set, with the model trained to reproduce daily or even monthly averaged species concentrations.

## 5   Conclusions

We have shown that a suitably trained machine learning based approach can replace the integration step within an atmospheric

chemistry model run on the timescale of days to weeks. The application of some chemical intuition, by which we separate long lived from short lived species, and a basic application of conservation of atoms to the $NO_x$ family, leads to significant improvements of model performance. The machine learning implementation is slower than the current model, but very little optimisation and software development has been thus far applied to the code.

Methodologies similar to this may offer the potential to accelerate the calculation of chemistry for some atmospheric chem-

istry applications such as ensembles of air quality forecasts and data assimilation. Future work on both the algorithm and the methodology is necessary to produce a useful solution but this first step shows promise.

*Code and data availability.* The GEOS-Chem model output used for training and validation will be made available upon final acceptance via the data repository of University York. A copy of the random forest training code (written in Python) and the model emulator (Fortran) is available upon request from Christoph Keller. GEOS-Chem (http://geos-chem.org) is freely available through an open license (http://acmg.

seas.harvard.edu/geos/geos_licensing.html). The GEOS-5 global modeling system is available through the NASA Open Source Agreement, Version 1.1 and can be accessed at https://gmao.gsfc.nasa.gov/GEOS_systems/geos5_access.php with further instruction available at https://geos5.org/wiki/index.php?title=GEOS-5_public_AGCM_Documentation_and_Access.

*Author contributions.* MJE and CAK came up with the concept and together wrote the paper. MJE developed the algorithm and CAK implemented it into the GEOS model. Both authors devised the experiments.

*Competing interests.* The authors declare no competing interests.

*Acknowledgements.* CAK acknowledges support by the NASA Modeling, Analysis, and Prediction (MAP) Program. Resources supporting the model simulations were provided by the NASA Center for Climate Simulation at Goddard Space Flight Center (https://www.nccs.nasa.gov/services/discover). MJE acknowledges support from the UK Natural Environment Research Council from the MAGNIFY and BACCUS grants (NE/M013448/1 and NE/L01291X/1). The authors thank J. Zhuang, M.M. Kelp, C W. Tessum, J.N. Kutz and N.D. Brenowitz for valuable discussion.

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

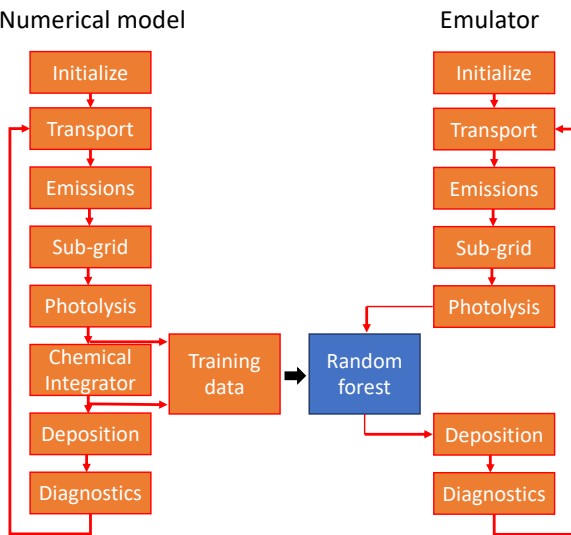

**Figure 1.** Schematic overview of the use of a random forest regression algorithm as an alternative to the chemistry solver. The original numerical model (GEOS-Chem) sequentially solves the operations relevant to atmospheric chemistry, with the chemical integrator being the computationally most expensive step (left side). Using training data produced from the full model, we generate a machine learning emulator that can then be used instead of the chemical integrator (right side). All other model processes are the same as in the original model.

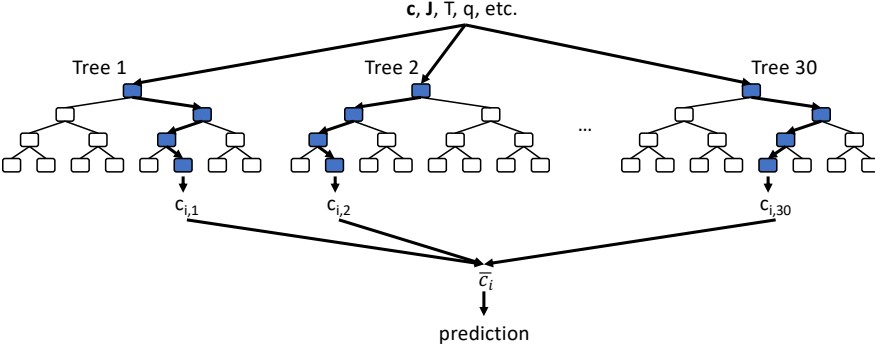

**Figure 2.** Schematic of random forest algorithm. For each species $c_i$, we use a random forest consisting of 30 individual decision trees, each up to 12 layers deep (only first four layers are shown). All decision trees take the same inputs (e.g. species concentration vector **c** at given location, photolysis rates **J**, temperature T, humidity q) and each decision tree node uses one of the input features plus a threshold value to determine the tree path for the given set of input features. The final prediction is made by averaging the 30 individual tree predictions ($c_{i,1}$, $c_{i,2}$, ..., $c_{i,30}$).

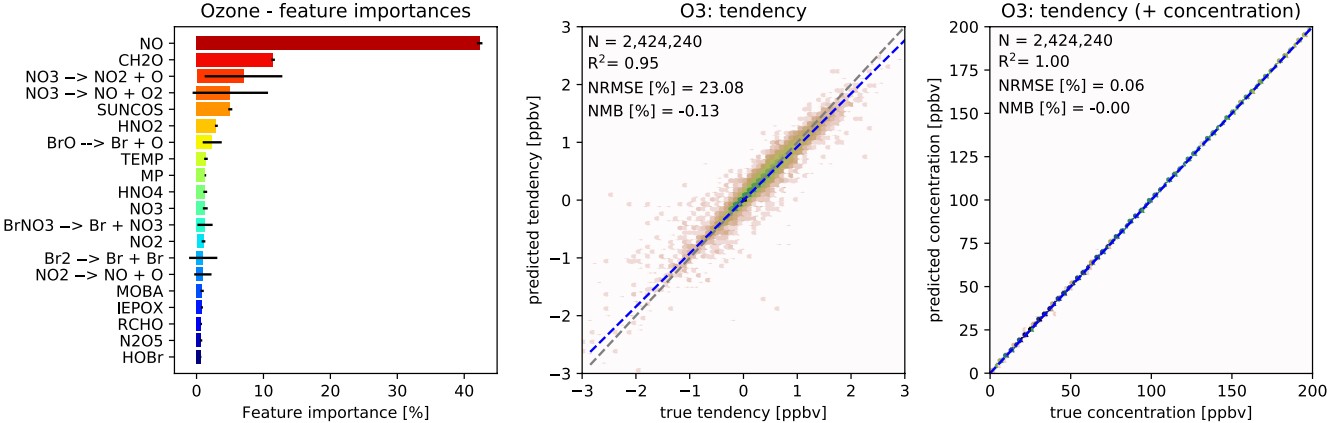

**Figure 3.** Characteristics of random forest trained to predict tendencies of $O_3$ due to chemistry. (Left) Importance of input variables (features) for random forests trained to predict tendency of ozone due to chemistry. Shown are the 20 most important features for the entire random forest, as averaged over all 30 decision trees. The black bars indicate the standard deviation for each feature across the 30 decision trees. The arrows indicate photolytic conversion (i.e. $NO_3$ photolyses to $NO_2$ plus O); (Middle) Validation of random forest prediction skill for ozone: comparison of ozone tendency validation data (x-axis) vs. predicted values (y-axis). Number of validation points (N), correlation coefficient ($R^2$), normalized root mean square error (NRMSE) and normalized mean bias (NMB) are given in the inset; (Right) Same validation but with tendency added to the concentration before integration.

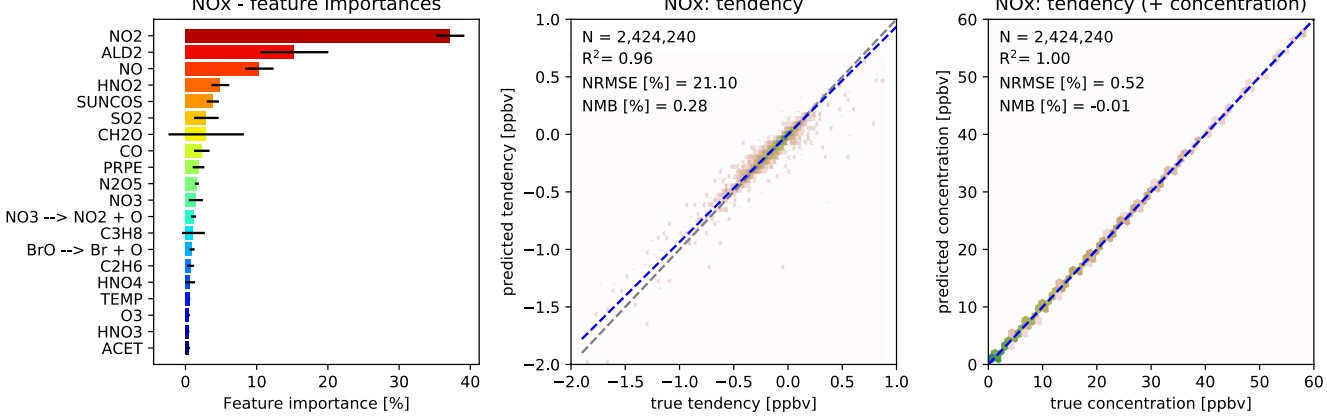

**Figure 4.** As Figure 3 but for $NO_x$ ($NO + NO_2$).

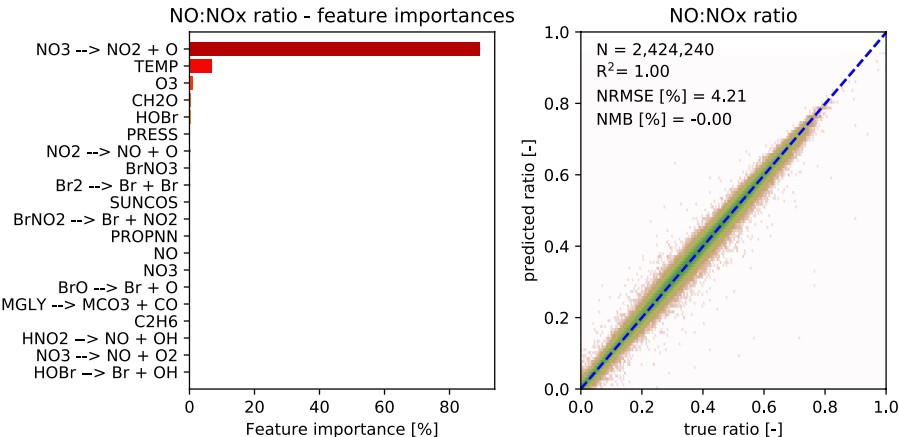

**Figure 5.** Characteristics of random forest trained to predict the NO/NO$_x$ ratio after chemistry. (Left) 20 most important features for the NO/NO$_x$ random forest, as averaged over all 30 decision trees. The black bars indicate the standard deviation of the feature importances; (Right) Comparison of predicted NO/NO$_x$ ratios (y-axis) vs. true NO/NO$_x$ ratios (x-axis) for the validation data (not used for training). Number of validation points (N), correlation coefficient (R$^2$), normalized root mean square error (NRMSE) and normalized mean bias (NMB) are given in the inset.

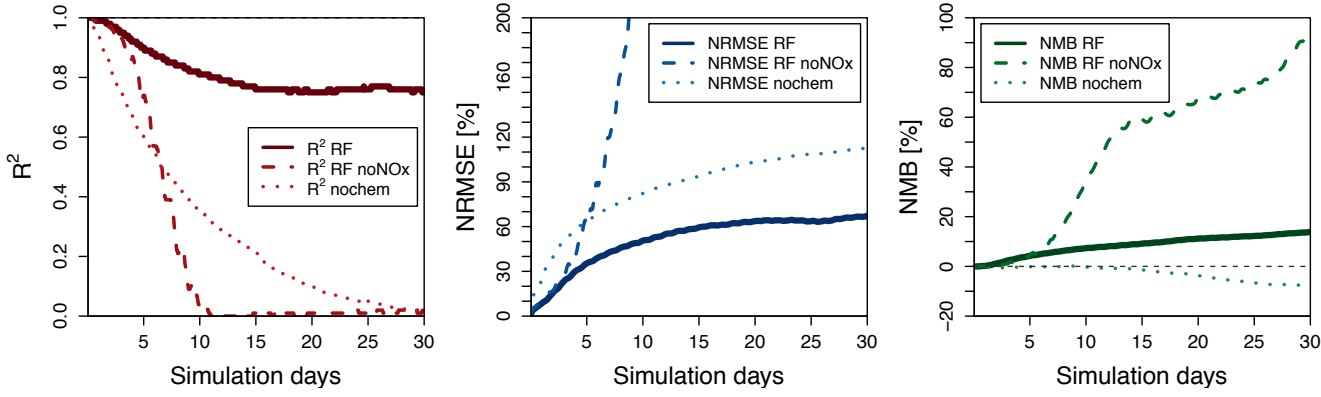

**Figure 6.** 30-day evolution of R$^2$ (left), NRMSE (middle), and NMB (right) for three different model simulations of O$_3$ run for July 2014 compared to full GEOS-Chem simulation. Solid line represents the standard RFR simulation using the family prediction of NO$_x$. Dashed line uses RFR predictors for NO and NO$_2$ individually (this simulation becomes unstable after 23 days). The dotted line represents a simulation with no chemistry. Grey line on the right hand plot indicates a 0 value.

## O3 surface concentrations

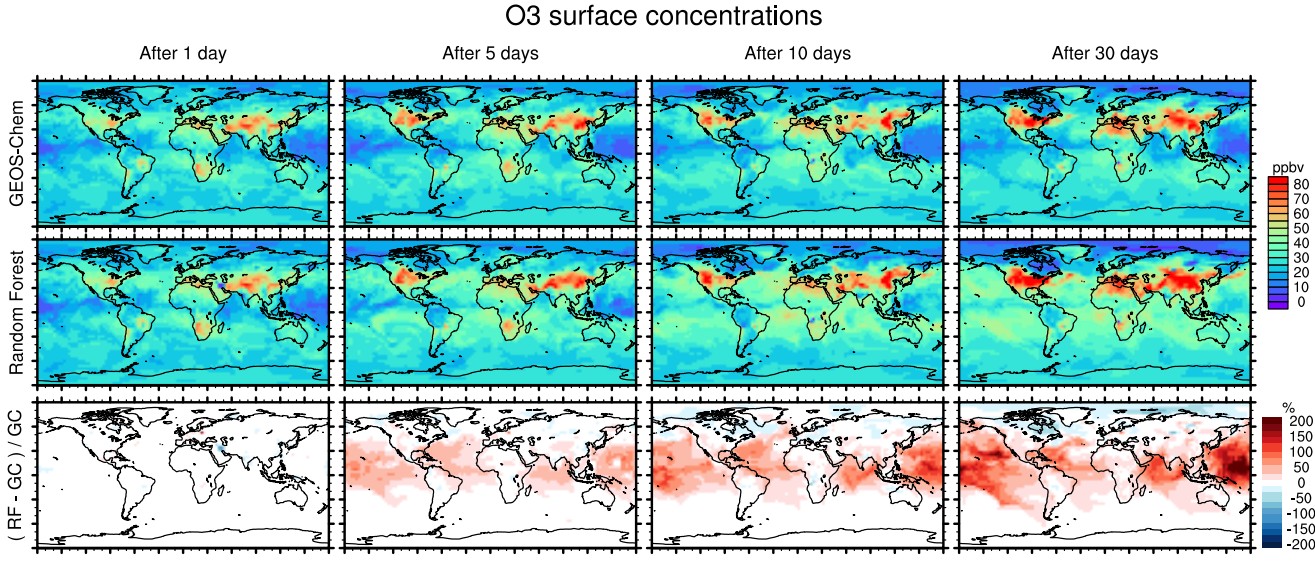

**Figure 7.** Concentration maps of surface $O_3$ mixing ratio after 1 simulation day (column 1), 5 simulation days (column 2), 10 simulation days (column 3), and 30 simulation days (column 4), as calculated by the full GEOS-Chem model (row 1) and the standard RFR model with the $NO_x$ family treatment (row 2). Row 3 shows the percentage difference between the RFR simulation and GEOS-Chem (GC).

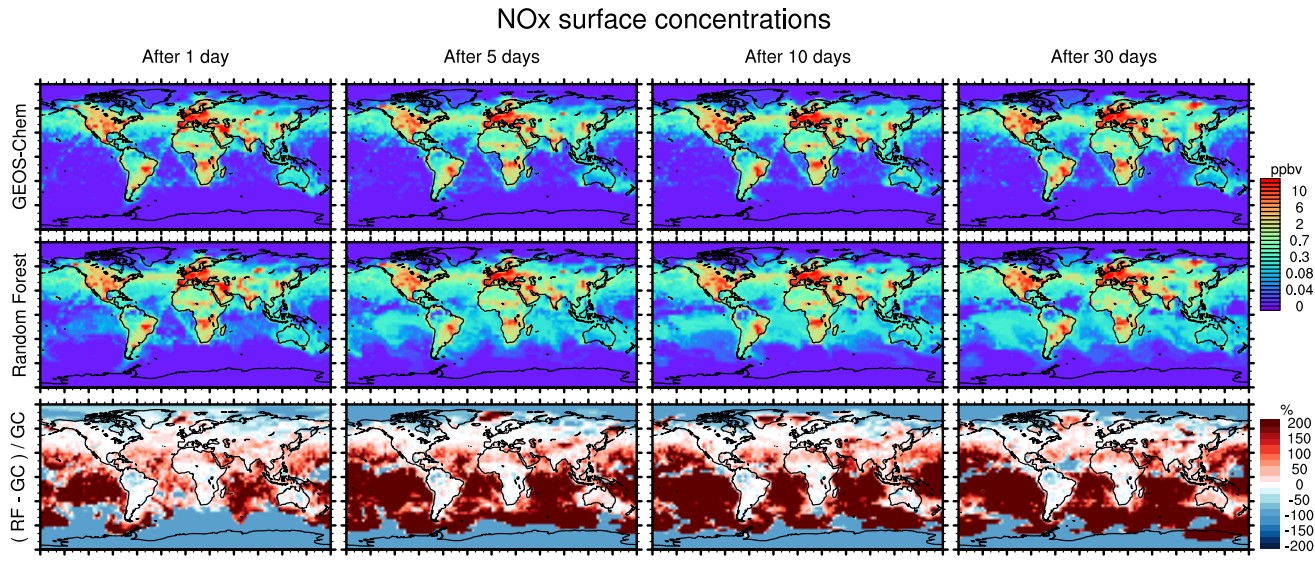

**Figure 8.** Concentration maps of surface $NO_x$ ($NO + NO_2$) after 1 simulation day (column 1), 5 simulation days (column 2), 10 simulation days (column 3), and 30 simulation days (column 4), as calculated by the full GEOS-Chem model (row 1) and the standard RFR model with the $NO_x$ family treatment (row 2). Row 3 shows the relative difference between the RFR simulation and GEOS-Chem (GC).

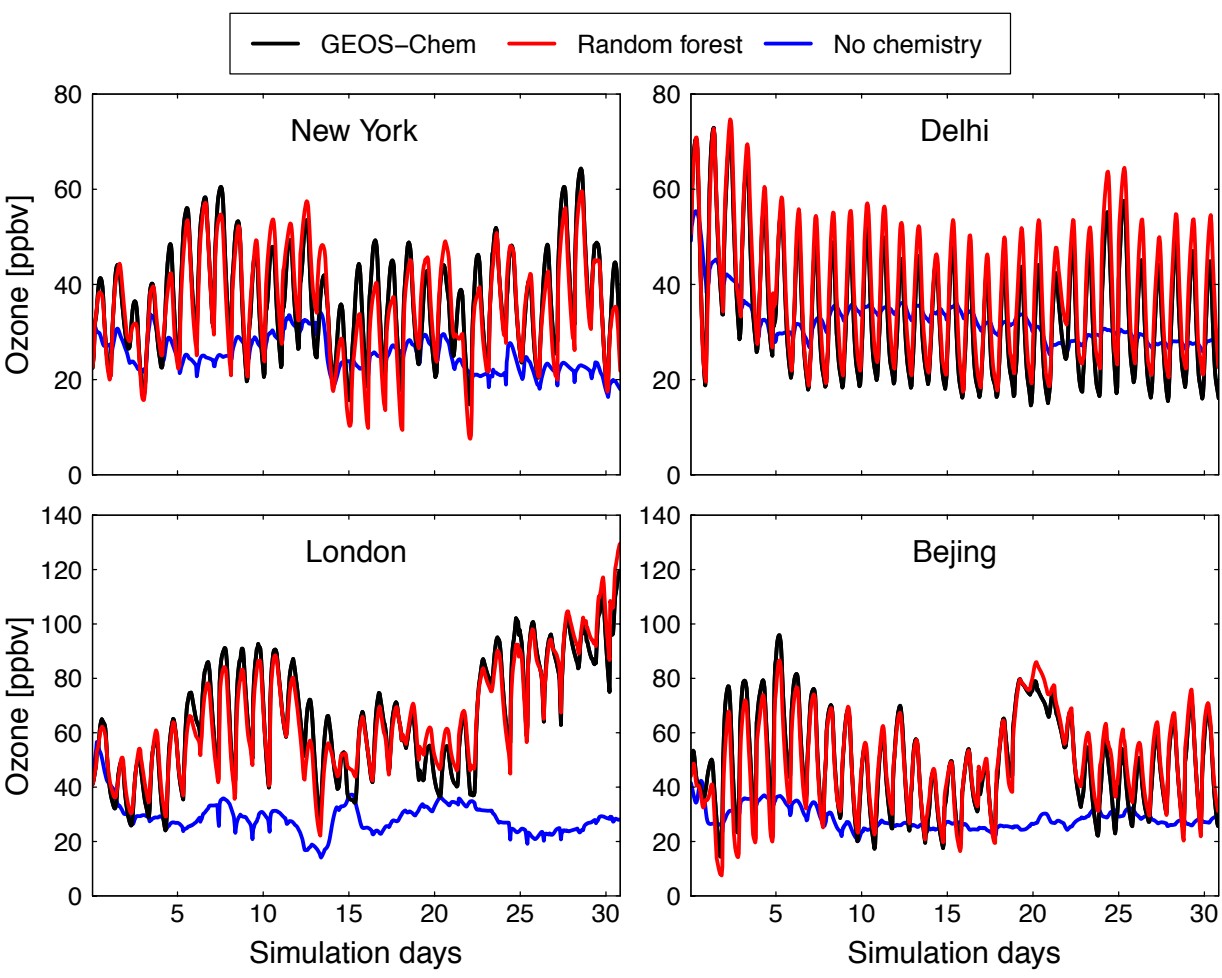

**Figure 9.** Comparison of surface concentration of $O_3$ at four locations (New York, Delhi, London and Bejing) for the GEOS-Chem reference simulation (black), the RFR model with the $NO_3$ family treatment (red) and a simulation with no chemistry (blue).

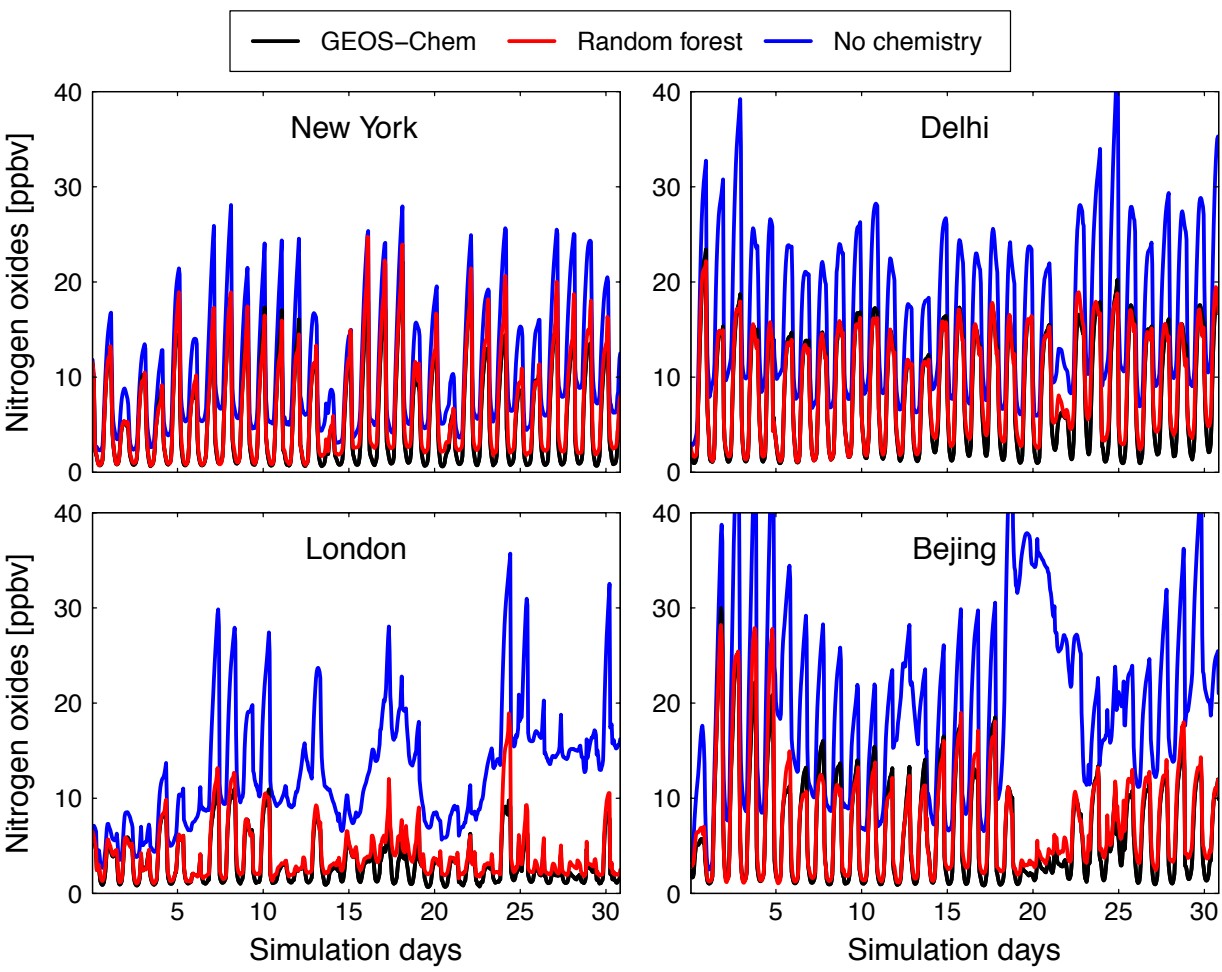

**Figure 10.** Comparison of surface concentration of nitrogen oxides ($NO_x$ = NO + $NO_2$) at four locations (New York, Delhi, London and Bejing) for the GEOS-Chem reference simulation (black), the RFR model with the $NO_3$ family treatment (red) and a simulation with no chemistry (blue).

**Table 1.** Overview of the performance of the RFR model with the $NO_x$ family treatment. Shown are the Pearson correlation $R^2$, normalized root mean square error (NRMSE), and normalized mean bias (NMB). Comparison against the validation data set (10% of training data withhold from training) are indicated with a 'V'. Comparisons between the RFR simulation and the full GEOS-Chem model for July 2014 at 00UTC after the 1st, 5th and 30st simulation day are indicated with 'D1', 'D5', and 'D30' respectively. Prediction type of each species (concentration, tendency, NOx family treatment) is given in the Prediction column.

| Nr | ID | Name | Prediction | $R^2$ | | | | NRMSE [%] | | | | NMB [%] | | | |
|---|---|---|---|---|---|---|---|---|---|---|---|---|---|---|---|
| | | | | V | D1 | D5 | D30 | V | D1 | D5 | D30 | V | D1 | D5 | D30 |
| 1 | ACET | Acetone | Tend | 0.98 | 1 | 1 | 1 | 15 | 0.88 | 2.3 | 3.7 | -0.29 | -0.039 | -0.39 | 0.17 |
| 2 | ALD2 | Acetaldehyde | Tend | 0.93 | 0.99 | 0.98 | 0.93 | 26 | 12 | 16 | 27 | -0.83 | -0.082 | -6.4 | 3.7 |
| 3 | ALK4 | ≥C4 alkanes | Tend | 0.98 | 1 | 1 | 1 | 13 | 2.6 | 5.8 | 7.6 | 0.06 | -0.12 | -0.074 | -11 |
| 4 | Br | Atomic bromine | Conc | 0.82 | 0.26 | 0.18 | 0.063 | 45 | 130 | 250 | 410 | -1.4 | 73 | 120 | 170 |
| 5 | Br2 | Molecular bromine | Conc | 0.84 | 0.87 | 0.84 | 0.47 | 40 | 38 | 49 | 82 | -6.4 | -18 | -30 | -42 |
| 6 | BrNO2 | Nitryl bromide | Conc | 0.87 | 0.82 | 0.84 | 0.76 | 40 | 46 | 45 | 57 | 8.5 | 46 | 54 | 39 |
| 7 | BrNO3 | Bromine nitrate | Conc | 0.42 | 0.33 | 0.4 | 0.42 | 110 | 150 | 140 | 140 | 110 | 190 | 170 | 160 |
| 8 | BrO | Bromine monoxide | Conc | 0.83 | 0.48 | 0.29 | 0.05 | 47 | 73 | 110 | 250 | -18 | 23 | 52 | 120 |
| 9 | C2H6 | Ethane | Tend | 0.98 | 1 | 1 | 1 | 13 | 1.4 | 4.8 | 9.1 | 0.0082 | -0.052 | -1.1 | -6.1 |
| 10 | C3H8 | Propane | Tend | 0.97 | 1 | 1 | 0.98 | 17 | 4.1 | 5.1 | 15 | -0.05 | 0.85 | 0.8 | -14 |
| 11 | CH2Br2 | Dibromomethane | Tend | 0.97 | 1 | 1 | 1 | 19 | 0.86 | 2.9 | 7.1 | -0.26 | 0.0036 | -0.24 | -1.8 |
| 12 | CH2O | Formaldehyde | Tend | 0.93 | 0.97 | 0.95 | 0.95 | 26 | 17 | 24 | 27 | -0.28 | 3.4 | 17 | 12 |
| 13 | CH3Br | Methyl bromide | Tend | 0.97 | 1 | 1 | 1 | 17 | 0.26 | 0.97 | 1.8 | -0.16 | 0.0013 | -0.033 | -0.044 |
| 14 | CHBr3 | Bromoform | Tend | 0.99 | 1 | 1 | 1 | 8.4 | 0.86 | 2.3 | 4.1 | -0.18 | -0.022 | -0.36 | -1.7 |
| 15 | CO | Carbon monoxide | Tend | 0.98 | 1 | 1 | 1 | 13 | 0.89 | 2.2 | 2.4 | 0.09 | 0.017 | -0.12 | -0.98 |
| 16 | DMS | Dimethylsulfide | Tend | 0.98 | 0.99 | 0.89 | 0.87 | 12 | 11 | 38 | 58 | -0.17 | -6.8 | -31 | -54 |
| 17 | GLYC | Glycoaldehyde | Tend | 0.97 | 0.99 | 0.99 | 0.98 | 17 | 11 | 14 | 16 | -0.30 | -5.5 | -8.1 | -8.5 |
| 18 | H2O2 | Hydrogen peroxide | Tend | 0.96 | 0.97 | 0.91 | 0.86 | 20 | 19 | 31 | 45 | 0.1 | -6 | -4.2 | 3.5 |
| 19 | HAC | Hydroxyacetone | Tend | 1 | 0.99 | 0.99 | 0.98 | 0.95 | 8.4 | 15 | 16 | 0.025 | -1.4 | -6.2 | -10 |
| 20 | HBr | Hydrobromic acid | Conc | 0.68 | 0.74 | 0.72 | 0.6 | 56 | 52 | 53 | 66 | 1.7 | 9.8 | 8.9 | 19 |
| 21 | HNO2 | Nitrous acid | Conc | 0.91 | 0.85 | 0.96 | 0.76 | 34 | 48 | 43 | 64 | -7.4 | 23 | 37 | 50 |
| 22 | HNO3 | Nitric acid | Conc | 0.88 | 0.88 | 0.87 | 0.77 | 37 | 36 | 39 | 55 | 2.3 | 12 | 27 | 37 |
| 23 | HNO4 | Peroxynitric acid | Conc | 0.71 | 0.72 | 0.74 | 0.69 | 55 | 60 | 56 | 64 | 4.2 | 40 | 50 | 65 |
| 24 | HOBr | Hypobromous acid | Conc | 0.7 | 0.59 | 0.54 | 0.47 | 57 | 73 | 73 | 86 | 12 | 23 | 16 | 28 |
| 25 | IEPOX | Isoprene epoxide | Tend | 0.98 | 0.98 | 0.97 | 0.97 | 15 | 17 | 21 | 19 | 0.06 | -4.1 | -5.2 | -5.8 |
| 26 | ISOP | Isoprene | Tend | 0.99 | 0.94 | 0.93 | 0.88 | 12 | 31 | 31 | 38 | -0.20 | -15 | -21 | -27 |

**Table 2.** Continuation of Table 1

| Nr | ID | Name | Prediction | R² | | | | NRMSE [%] | | | | NMB [%] | | | |
|---|---|---|---|---|---|---|---|---|---|---|---|---|---|---|---|
| | | | | V | D1 | D5 | D30 | V | D1 | D5 | D30 | V | D1 | D5 | D30 |
| 27 | ISOPN | Isoprene nitrate | Tend | 0.94 | 0.94 | 0.92 | 0.78 | 24 | 28 | 30 | 48 | -3.0 | -19 | -18 | -14 |
| 28 | MACR | Mathacrolein | Tend | 0.97 | 0.98 | 0.96 | 0.88 | 17 | 18 | 27 | 38 | 2.3 | -12 | -21 | -28 |
| 29 | MAP | Peroxyacetic acid | Tend | 0.96 | 0.99 | 0.98 | 0.98 | 20 | 8.6 | 17 | 15 | -0.29 | -2 | -6.8 | 0.27 |
| 30 | MEK | Methyl ethyl ketone | Tend | 0.91 | 0.98 | 0.98 | 0.96 | 31 | 15 | 14 | 25 | -0.73 | -0.39 | -0.22 | 28 |
| 31 | MMN | MACR + MVK nitrate | Tend | 0.97 | 0.98 | 0.95 | 0.89 | 17 | 14 | 22 | 38 | 0.61 | -2.9 | -7 | -5.1 |
| 32 | MOBA | 5C acid from isoprene | Conc | 0.98 | 0.95 | 0.93 | 0.87 | 15 | 25 | 29 | 37 | -2.8 | -14 | -16 | -18 |
| 33 | MP | Methylhydroperoxide | Tend | 0.89 | 0.97 | 0.8 | 0.8 | 33 | 19 | 54 | 48 | -0.68 | -4.6 | -19 | -15 |
| 34 | MPN | Methyl peroxy nitrate | Conc | 0.85 | 0.62 | 0.4 | 0.43 | 50 | 87 | 130 | 140 | 26 | 100 | 160 | 130 |
| 35 | MSA | Methanesulfonic acid | Tend | 0.99 | 0.99 | 0.97 | 0.92 | 11 | 9.4 | 19 | 34 | -0.26 | -0.75 | -8.9 | -30 |
| 36 | MVK | Methylvinylketone | Tend | 0.96 | 0.98 | 0.96 | 0.83 | 19 | 17 | 27 | 42 | 1.3 | -9.9 | -21 | -27 |
| 37 | N2O5 | Dinitrogen pentoxide | Conc | 0.69 | 0.02 | 0.02 | 0.041 | 56 | 390 | 490 | 340 | 28 | 1700 | 2400 | 1800 |
| 38 | NO | Nitric oxide | NOx tend | 0.95 | 0.89 | 0.86 | 0.79 | 26 | 34 | 40 | 47 | -1 | 23 | 31 | 17 |
| 39 | NO2 | Nitrogen dioxide | NOx tend | 0.94 | 0.9 | 0.9 | 0.91 | 28 | 34 | 33 | 31 | 2.2 | 19 | 28 | 29 |
| 40 | NO3 | Nitrate radical | Conc | 0.74 | 0.064 | 0.065 | 0.095 | 60 | 690 | 620 | 470 | 30 | 780 | 840 | 850 |
| 41 | O3 | Ozone | Tend | 0.95 | 0.99 | 0.9 | 0.75 | 23 | 8.3 | 35 | 67 | -0.13 | 0.19 | 4.2 | 13 |
| 42 | PAN | Peroxyacetylnitrate | Tend | 0.91 | 0.95 | 0.89 | 0.77 | 30 | 22 | 35 | 59 | -4.8 | 1.3 | 8.3 | 23 |
| 43 | PMN | Peroxymethacroyl nitrate | Tend | 0.86 | 0.92 | 0.89 | 0.86 | 38 | 36 | 46 | 47 | -2.6 | 19 | 33 | 32 |
| 44 | PPN | Peroxypropionyl nitrate | Tend | 0.92 | 0.95 | 0.91 | 0.36 | 29 | 24 | 32 | 610 | -8.0 | 1.9 | 10 | 700 |
| 45 | PROPNN | Propanone nitrate | Tend | 0.89 | 0.99 | 0.97 | 0.97 | 33 | 11 | 17 | 31 | 0.05 | -0.28 | -2.2 | 9.8 |
| 46 | PRPE | ≥C3 alkenes | Tend | 0.96 | 0.99 | 0.95 | 0.88 | 20 | 11 | 22 | 36 | -0.23 | -5.2 | -11 | -15 |
| 47 | R4N2 | ≥C4 alkylnitrates | Tend | 0.88 | 0.94 | 0.94 | 0.84 | 35 | 26 | 27 | 90 | -0.83 | 2.4 | 7.4 | 60 |
| 48 | RCHO | ≥C3 aldehydes | Tend | 0.85 | 0.95 | 0.89 | 0.0 | 39 | 23 | 35 | 4900 | 1.3 | -0.71 | 4.1 | 13000 |
| 49 | RIP | Peroxide from RIO2 | Tend | 0.97 | 0.95 | 0.94 | 0.95 | 17 | 24 | 27 | 23 | -0.55 | -4.8 | -8.1 | -7.7 |
| 50 | SO2 | Sulfur dioxide | 0.99 | 1 | 1 | 1 | 12 | 0.49 | 1.3 | 2.9 | 8.6 | 0.53 | 0.79 | -1.7 | -7.6 |
| 51 | SO4 | Sulfate | Tend | 0.99 | 1 | 0.99 | 0.95 | 12 | 6.4 | 9.3 | 23 | 0.03 | -0.48 | 0.34 | 2.3 |
| 52 | NOx | NO + NO2 | Tend | 0.96 | 0.98 | 0.98 | 0.95 | 21 | 14 | 16 | 22 | 0.28 | 20 | 28 | 26 |