# Peer review of "Application of random forest regression to the calculation of gas-phase chemistry within the GEOS-Chem chemistry model v10"

_Geoscientific Model Development, 2018_

## Short Comment (SC1) · 27 Nov 2018

This is a highly interesting paper, which I enjoyed reading. Just a few comments that I would recommend considering/discussing in a revised version:

1. Using random forests, the authors here focus on emulating the chemical system of an air pollution forecasting model for the troposphere. However, machine learning has also been used to forecast air pollution itself (e.g. Mallet et al. 2008).

[Figure]

2. Composition changes such as ozone are also important for climate, which is a topic not considered here, but worth mentioning in the Discussions (cf. Nowack et al. 2018). I assume due to the use of a chemistry-transport model you did not consider feedbacks of ozone on meteorology/climate? It would be good to briefly contrast the challenges in air pollution modelling as compared to climate modelling where stratospheric ozone changes are particularly important and still expensive to calculate (cf. Nowack et al. 2015).

3. Concerning the selection of the cross-validation method: since you predict time series of chemical species concentrations/concentration changes, the samples for longer-lived species are not independent. From the current description in the paper, it seems that these time-dependencies were not taken into account in the cross-validation. Are the authors using a sequential cross-validation method, e.g.

   http://scikit-learn.org/stable/modules/generated/sklearn.model_selection. TimeSeriesSplit.html

   which was also used and described in Nowack et al. (2018)?

4. How is the boundary between the troposphere and stratosphere (where chemistry is interactive but linearized) handled? Is there any effect of the tropospheric machine learning predictions on the stratosphere? If yes, could this in turn affect some of the tropospheric results, for example, due to changes in the photochemical environment, or transport?

**References:**

1. Mallet V., Stoltz G., Mauricette B. Ozone ensemble forecast with machine learning algorithms. Journal of Geophysical Research: Atmospheres 114, D05307, doi:10.1029/2008JD009978, (2009).

[Figure]

2. Nowack P., Braesicke P., Haigh J., Abraham N. L., Pyle J., Voulgarakis A. Using machine learning to build temperature-based ozone parameterizations for climate sensitivity simulations. Environmental Research Letters 13, 104016, doi:10.1088/1748-9326/aae2b, (2018).

3. Nowack P. J., Abraham N. L., Maycock A. C., Braesicke P., Gregory J. M., Joshi M. M., Osprey A., Pyle J. A. A large ozone-circulation feedback and its implications for global warming assessments. Nature Climate Change, 5(1), 41–45. doi:10.1038/nclimate2451, (2015).

---

## Referee Comment (RC1) · Anonymous Referee #1 · 11 Dec 2018

Review of

Application of random forest regression to the calculation of gas-phase chemistry within the GEOS-Chem chemistry model v10 by Keller and Evans

Overview

The paper reports on the application of a machine learning (ML) approach (random forest regression, RFR) to replace the calculation of the chemical mechanism in the GEOS-Chem chemistry model. The RFR technique is able to reproduce the standard

application of GEOS-Chem for a 1-month simulation with a good degree of agreement. No computational optimisation has been carried, in fact the presented ML approach increases the cost by a factor 2.

General remarks

ML approaches are more and more tested to improve computational performance of Earth-System models. As the paper seems to be the first application of ML in a global CTM the therefore pioneering effort fully justifies publication in GMD. However, a revised manuscript should provide more detail on the following two main points and the more specific point raised in the specific remarks.

1) More detailed discussion of the motivation, limitations and prospect of the ML approach.

2) More detailed explanation of the RFR technique and its implementation of the model.

to 1)

ML is used to reproduce relations between input and output data sets for which there is (i) no quantitative understanding of the processes,i.e. no "model", or (ii) to reproduce complex model simulations in a cost efficient way. I think the current paper belongs more to the latter category. Therefore the reported increase in cost is not yet an indication that ML will find a place in CTM modelling. Avenues for cost saving of CTM simulations could be discussed with more concrete detail, especially approaches to optimise the application of chemical mechanisms by using reduced schemes in area away from the polluted regimes.

Apart from forecast applications, CTMs are often tools to study the response to variations in emissions on atmospheric composition. It is therefore an important question to know to what extent the accuracy of the ML scheme, i.e. good agreement with the standard CTM, can be maintained without the need to retrain the RF when emission change within a certain range. An actual evaluation of such a test (say 10% reduction

in emissions) could be a welcome extension of the paper.

to 2) A common GMD reader (as I am) may not be familiar with the RFR technique, which is why an extended explanation of the approach would be welcome. Further, the implementation in the CTM should be explained in more detail. For example, it was not always clear if the RFR is trained for all grid points, i.e. composition regimes, together or if a spatial or temporal stratification was applied and specifically trained algorithm would be called for each grid point.

Specific comments

P1 L12: please clarify that "error" is not error w.r.t observations

P1 L20: "85% slower" is not completely clear. Better say factor 2 more expensive or increase in time.

P2 L5: Provide unit of species to check consistency of formulae (density). Consider adding diffusion term if the equation is meant to refers to grid box mean.

P3 L24: As photolysis rate computations can be costly, please explain, why photolysis rates were not subject to the ML approach. They might be a good candidate too.

P3 L25: Please clarify what "each training data set" means - each model grid point ?

P3 L26/27: It is not clear what the numbers in brackets represent (space and time dimensions ?). The number given for the space and time gridpoints of a CTM simulation is not surprising. Please clarify if these numbers are untypical or a challenge to the development of RFR.

P4 L5: please explain the underlying idea of RF in a couple of sentences

P4L13: Why 30 trees ?

P4 L17: What is the computational cost / time to build the forest?

P5 L10: Please clarify what the "->" terms mean in Figure 1, the respective photolysis

rates ?

P5 L11: Why does O3 itself does not appear in the feature importance list? Why is NO3 photolysis more important than NO2 photolysis for ozone?

P5 L14: Does this refer to the troposphere as a whole or to specific regions?

P5 L17: I am not sure "misleading" is the right word here. It is just a manifestation of the fact that relative ozone tendencies are small.

P5 L31: Explain the impact of SO2 (as aerosol precursor ?)

P8 L20: Why is there no stronger increase in NOx when no chemical loss occurs in Figure 8?

P9 L17: I would argue this is exactly the same of a chemical mechanism. There is no dependency between grid point especially as column depends (ozone) of the photolysis calculation has been taken out of the equation. Please clarify, why this is a unique feature of the ML approach.

P9 L20: Please comment on memory cost of the RFR implementation. The demands for floating point operation and memory determines the type of suitable architecture.

P10 L5: I think this point has a large potential, i.e. to select the complexity of the chemical mechanism according to location. In remote areas less demanding chemistry calculation might be sufficient, which are the majority of the grid points. (There might be issues with latency in the parallelism if specific points have more complex schemes to solve)

P10 L22: Please specify to what extent the current RFR and standard implementation conserves the stochimetry.

P11 L4: Please provide more detail why this is easy and would be successful as small changes also accumulate over time.

P11 L22: Please quantify in more detail what amount of changes in emissions would be acceptable without need to retrain the RFR.

P11 L26: Despite the computational cost there is a fundamental reason why ensemble approaches do not have the same importance in AQ forecast than in weather forecasting, namely the less chaotic dependence of AQ forecast on AQ initial conditions. But I agree that significantly cheap AC simulations would be of great benefit. For example, high resolution NWP model runs could afford to have AC modules embedded, which could be used to simulate the impact of aerosols and chemistry on the weather.

---

## Author Comment (AC1) · 16 Jan 2019

**Response to interactive comment by P. Nowack**

We thank Peer Nowack for his valuable comments. Please find below our comments.

*1. Using random forests, the authors here focus on emulating the chemical system of an air pollution forecasting model for the troposphere. However, machine learning has also been used to forecast air pollution itself (e.g. Mallet et al. 2008).*
There are many potential applications for machine learning in atmospheric chemistry modeling, and this paper highlights one use case. We chose to focus on the emulation of chemical kinetics as this process is the computational bottleneck in chemistry transport models. However, as pointed out, alternative approaches could be to directly predict air pollution of selected target species, and/or to emulate the entire modeling system (transport, diffusion, emission & deposition, chemistry). We extended the introduction/discussion as follows to better highlight our motivation and other possible approaches:

"Machine learning emulators have also been explored to directly predict air pollution concentration in future time steps (Mallet et al., 2009), as well as for chemistry-climate simulations focusing on model predictions of time-averaged concentrations for selected species such as ozone and OH over time scales of days to months (Nicely et al., 2017; Nowack et al., 2018). In contrast, the algorithm presented here is optimised to capture the small-scale variability of the entire chemical space within a time scale of minutes, with only a small loss of accuracy when used repeatedly over multiple time steps."

"Another potential application area for machine-learning based chemistry emulators are chemistry-climate simulations. Unlike air quality applications, which focus on small-scale variations of air pollutants over comparatively short periods of time of days to weeks, chemistry-climate studies require long simulation windows of the order of decades. Because of this, machine learning models used for these applications need to be optimised such that they accurately reproduce the (long-term) response of selected species - e.g. ozone and OH - to key drivers such as temperature, photolysis rates and NOx (Nicely et al., 2017; Nowack et al., 2018). The here presented method could be optimised for such an application by simplifying the problem set, with the model trained to reproduce daily or even monthly averaged species concentrations."

*2. Composition changes such as ozone are also important for climate, which is a topic not considered here, but worth mentioning in the Discussions (cf. Nowack et al. 2018). I assume due to the use of a chemistry-transport model you did not consider feedbacks of ozone on meteorology/climate? It would be good to briefly contrast the challenges in air pollution modelling as compared to climate modelling where stratospheric ozone changes are particularly important and still expensive to calculate (cf. Nowack et al. 2015).*
We indeed ignore here the feedbacks of atmospheric constituents on weather and climate since we use a chemical transport model where the meteorology is fixed. In addition, our focus is on the accurate representation of chemistry at high temporal resolution (hours) but over a relatively short period of time (weeks to months). For chemistry-climate applications, one would need to prioritize the long-term stability and chemical balance of the machine learning system over the short-term accuracy of local surface concentrations. We discuss this now in more detail in the discussions, as already highlighted above. In addition, we added the following sentence to the model description:

"While the GEOS model with GEOS-Chem chemistry can be run as a chemistry-climate model where the chemical constituents (notably ozone and aerosols) directly feed back to the meteorology, we

disable this option here and use prescribed ozone and aerosol concentrations for the meteorology instead. This ensures that any differences between the reference model and the machine learning model can be attributed to imperfections in the emulator, rather than changes in meteorology due to chemistry-climate feedbacks."

*3. Concerning the selection of the cross-validation method: since you predict time series of chemical species concentrations/concentration changes, the samples for longer-lived species are not independent. From the current description in the paper, it seems that these time-dependencies were not taken into account in the cross-validation. Are the authors using a sequential cross-validation method, e.g.*
*http://scikit-learn.org/stable/modules/generated/sklearn.model_selection. TimeSeriesSplit.html*
*which was also used and described in Nowack et al. (2018)?*
We address the problem of time series correlation in two ways: (1) our emulator is time-agnostic, i.e. we focus on the prediction of concentrations / tendencies due to chemistry at time X based on the chemistry initial conditions at the same time (rather than e.g. predicting concentrations at X + Δt based upon concentrations at time X); (2) by predicting the change in concentrations for long-lived species instead of absolute concentrations, we reduce the temporal correlations between subsequent time steps. Having that said, we fully acknowledge that the training samples are not fully independent, and neither are the input variables. However, one of the advantages of the random forest algorithm is its relative robustness even when using correlated variables as long as the full sampling set is representative of the underlying process, which we consider to be the case for the time period considered in this study.

*4. How is the boundary between the troposphere and stratosphere (where chemistry is interactive but linearized) handled? Is there any effect of the tropospheric machine learning predictions on the stratosphere? If yes, could this in turn affect some of the tropospheric results, for example, due to changes in the photochemical environment, or transport?*
For our study we consider the impact from stratosphere-troposphere exchange (STE) to be small given the relatively short time window. We address this in more detail in section 3:

"In all simulations the stratospheric chemistry uses a linearised chemistry scheme (Murray et al., 2012). This buffers the impact of the RFR emulator over the long-term since all simulations use the same relaxation scheme in the stratosphere. For the here considered time frame of one month, we consider this impact to be negligible in the lowest 25 model levels."

---

## Author Comment (AC2) · 16 Jan 2019

**Response to Review #1**

We thank the reviewer for taking the time to provide a thorough review. Our responses to each comment/remark (in italics) are given below.

*General remarks*

*ML approaches are more and more tested to improve computational performance of Earth-System models. As the paper seems to be the first application of ML in a global CTM the therefore pioneering effort fully justifies publication in GMD. However, a revised manuscript should provide more detail on the following two main points and the more specific point raised in the specific remarks.*
*1) More detailed discussion of the motivation, limitations and prospect of the ML approach.*
*2) More detailed explanation of the RFR technique and its implementation of the model.*

*to 1) ML is used to reproduce relations between input and output data sets for which there is (i) no quantitative understanding of the processes, i.e. no "model", or (ii) to reproduce complex model simulations in a cost efficient way. I think the current paper belongs more to the latter category. Therefore the reported increase in cost is not yet an indication that ML will find a place in CTM modelling. Avenues for cost saving of CTM simulations could be discussed with more concrete detail, especially approaches to optimise the application of chemical mechanisms by using reduced schemes in area away from the polluted regimes.*

This paper shows that there are inherent low-rank structures in the chemical kinetics that can be exploited using machine learning algorithms. The current implementation should be seen as a proof of concept, rather than a fully developed model emulator. As also pointed out by the reviewer, there are many different possible ways to save computation time of a CTM, and exploring them in detail here is beyond the scope of this project. However, our work shows that random forests can reproduce the concentration changes computed by a numerical model with high accuracy. Given the embarrassingly parallel structure of the random forest algorithm, we are confident that much faster implementations are realistic, e.g. by evaluating the random forest on GPUs.

In the modified version of the manuscript we added Figure 1 to provide more detail on the implementation and further discuss ways to improve the efficiency, e.g. by changing from double-precision to single-precision or optimize the number of trees and/or leaves per trees.

*Apart from forecast applications, CTMs are often tools to study the response to variations in emissions on atmospheric composition. It is therefore an important question to know to what extent the accuracy of the ML scheme, i.e. good agreement with the standard CTM, can be maintained without the need to retrain the RF when emission change within a certain range. An actual evaluation of such a test (say 10% reduction in emissions) could be a welcome extension of the paper.*

The one-month simulation experiment for July 2014 was conducted using a slightly different set of emissions compared to the training data (due to changes in annual scale factors), with local emission differences of up to 20%. The experiment thus demonstrates that the RFR algorithm indeed captures the chemistry response to changes in emissions, at least for changes in emissions of the order of 10-20%. We expand on this in the revised version of the manuscript in section 3 and 4.5 by adding the following sentences:

"This simulation differs from the training simulation not only in meteorology but also in emissions, with local differences in NOx, CO, and VOC emissions of up to 20%. As such, this experiment also evaluates the ability of the RFR model to capture the sensitivity of chemistry to changes in emissions."

*to 2) A common GMD reader (as I am) may not be familiar with the RFR technique, which is why an extended explanation of the approach would be welcome. Further, the implementation in the CTM should be explained in more detail. For example, it was not always clear if the RFR is trained for all grid points, i.e. composition regimes, together or if a spatial or temporal stratification was applied and specifically trained algorithm would be called for each grid point.*
We extended the description of RFR in section 2.3 and also added a figure with a schematic overview of the algorithm:

"Figure 2 shows a schematic of RFR. It is a commonly used, and conceptually simple, supervised learning algorithm that consists of an ensemble (or forest) of decision trees. Each tree contains a tree-like sequence of decision nodes, based on which the tree splits into its various branches until the end of the tree ('the leaf') is reached. This leaf is the prediction of the decision tree. Each decision node is based on whether one of the input features is above a certain value. An important aspect of the random forest is that each tree of the forest is trained on a subset of the full training data, thus providing a slightly different approximation of the model. A prediction is then made by averaging the predictions of the individual trees."

We also updated section 2.4 to provide more detail on the implementation of the algorithm, including training time and use within the chemistry model:

"This predictor can be applied to all model grid cells, i.e. it captures all chemical regimes encountered by the respective target species. Conceptually, one can imagine that each tree path represents a different chemical regime, so it is important to generate trees that are large enough to encompass the entire solution space. We find a good compromise between computational complexity and accuracy of the solutions for random forests consisting of 30 trees with a maximum of 10,000 leaves (prediction values) per tree. These hyper-parameter were determined by trial-and-error, and we find very little sensitivity of our results to changes (+/-50%) to the number of trees and/or number of leaves.
…
We distributed the training of the entire forest (30 trees for 51 species) onto 1,530 CPU's, and each tree took one hour to train. "

**Specific comments**
*P1 L12: please clarify that "error" is not error w.r.t observations*
Corrected.

*P1 L20: "85% slower" is not completely clear. Better say factor 2 more expensive or increase in time.*
Corrected.

*P2 L5: Provide unit of species to check consistency of formulae (density). Consider adding diffusion term if the equation is meant to refers to grid box mean.*
We added the units for the species. Diffusion is negligibly slow in the troposphere and stratosphere

and hence we ignore it. We added the followoing sentence to clarify this:
"We ignore here molecular diffusion as it is negligibly slow compared to advection. "

*P3 L24: As photolysis rate computations can be costly, please explain, why photolysis rates were not subject to the ML approach. They might be a good candidate too.*
This is an excellent point and we did consider this. For example, one could develop an ML model where the computation of photolysis rates is embedded in the chemical ML emulator: instead of the photolysis rates, such a model would use as input parameter additional environmental fields relevant to photolysis, e.g. cloud cover, overhead ozone and aerosol loadings. We decided not to pursue this for this study because our main interest was to find out how close an ML model can emulate the chemistry, without additional possible errors e.g. from emulating other processes. We added a sentence for clarification:

"In addition, we output all photolyis rates since those are an essential element for chemistry calculations. Alternatively, one could also envision to directly embed the (computationally demanding) photolysis computation into the ML model, such that the emulator takes as input variables additional environmental variables relevant to photolysis (e.g. cloud cover, overhead ozone and aerosol loadings) and then emulates photolysis computation along with chemistry."

*P3 L25: Please clarify what "each training data set" means - each model grid point ?*
Thanks for pointing this out. We mean each model grid point. We changed the wording to clarify this.

*P3 L26/27: It is not clear what the numbers in brackets represent (space and time dimensions ?). The number given for the space and time gridpoints of a CTM simulation is not surprising. Please clarify if these numbers are untypical or a challenge to the development of RFR.*
We added the dimensions to the bracket. In addition, we elaborate more on the sensitivity of our results to the number of input samples.
The number of training samples is not a problem for the RF algorithm. As with all machine learning algorithms, the key is to train the model on a data set that is representative of the full solution space without overfitting. The random forest is a very robust machine learning algorithm and less prone to overfitting than other algorithms. Indeed, we find very little change to our results when lowering or increasing the number of samples used for training or excluding parts of the training set altogether (out of bag sampling) as long as the training set retains the full solution space.

*P4 L5: please explain the underlying idea of RF in a couple of sentences*
We updated the description of the RF, as already discussed above.

*P4L13: Why 30 trees?*
There is no generic way to define the hyper-parameter of the random forest model (number of trees, max. number of leaves, etc.). The values chosen here are based on manual trial-and-error. We find that our results are very robust to changing the hyper-parameter by +/- 50% (i.e. 15 or 45 trees instead of 30, etc.). This is now also included in the description:

"This predictor can be applied to all model grid cells, i.e. it captures all chemical regimes encountered by the respective target species. Conceptually, one can imagine that each tree path represents a different chemical regime, so it is important to generate trees that are large enough to encompass the entire solution space. We find a good compromise between computational complexity and accuracy of the solutions for random forests consisting of 30 trees with a maximum

of 10,000 leaves (prediction values) per tree. These hyper-parameter were determined by trial-and-error, and we find very little sensitivity of our results to changes (+/-50%) to the number of trees and/or number of leaves. "

*P4 L17: What is the computational cost / time to build the forest?*
The time to build the model depends on the hardware and software. The here used software package (scikit-learn) does not support Graphic Processing Units (GPUs), which adversely effects computation time. It took us one hour to build one tree, i.e. the entire forest (30 trees for 51 species) would take 1530 hours if trained on one CPU. We added this information to the manuscript, as already stated above.

*P5 L10: Please clarify what the "->" terms mean in Figure 1, the respective photolysis rates?*
Clarified.

*P5 L11: Why does O3 itself does not appear in the feature importance list? Why is NO3 photolysis more important than NO2 photolysis for ozone?*
Maybe somewhat counter-intuitively, ozone is not among the 20 most important input features for the prediction of ozone. The main reason for this is that we choose to predict the change of ozone (i.e. the production and loss), which is more sensitive to availability of NOx, VOCs, photolysis, etc., rather than the initial concentration of ozone. If the random forest model was trained to predict the absolute concentration of ozone, the initial ozone concentration would be the by far most important input feature (explaining more than 99% of the prediction).
As for the importance of NO3 photolysis over NO2 photolysis, we think that this is a somewhat 'random' result given the high correlation of the various photolysis rates. This is corroborated by the very high variability of photolysis rate feature importance across the 30 trees (black bars in left panel of Figure 1). We added the following sentence to the revised manuscript:

"For ozone prediction, 6 out of the 20 most important input features are related to photolysis. Most of the photolysis rates are highly correlated, and the individual decision trees use different photolysis rates for decision making. This results in very large standard deviations for the photolysis input features across the 30 decisison trees, as indicated by the black bars in the left panel of Figure 3.
Note that the concentration of O3 is not among the 20 most important input features for the prediction of O3 tendency. If, instead, the random forest model is trained to predict the concentration of O3, the initial O3 concentration dominates the input feature importance, explaining more than 99% of the prediction. However, when predicting the ozone tendency, the random forest algorithm is more sensitive to availability of NOx, VOCs, photolysis, etc., rather than the initial concentration of O3. For regions producing ozone (dominated by the NO+HO2−>NO2+OH reaction) the O3 concentration is not the primary source of variability. Similarly, for regions loosing ozone the dominant source of variability is the variability in photolysis rates (multiple orders of magnitude) rather than the variability in O3 concentration (less than an order of magnitude)."

*P5 L14: Does this refer to the troposphere as a whole or to specific regions?*
We look at the (lower to mid) troposphere as a whole, which we clarify in the updated version of the manuscript.

*P5 L17: I am not sure "misleading" is the right word here. It is just a manifestation of the fact that relative ozone tendencies are small.*
Agreed. We changed the wording.

*P5 L31: Explain the impact of SO2 (as aerosol precursor?)*
We think that SO2 serves as a proxy for aerosol abundance, picked up by the random forest as the determinant for heterogenous reactions in the absence of more detailed aerosol information. We modified the sentence as following:
"The importance of SO2 may reflect heterogeneous N2O5 chemistry, with SO2 being a proxy for available aerosol surface area (note that we do not provide any aerosol information to the RFR). "

*P8 L20: Why is there no stronger increase in NOx when no chemical loss occurs in Figure 8?*
We are still allowing for deposition processes, which especially at the surface act as a buffer to the increase from emissions. In addition, NOx emitted at the surface is lifted by convection into the free troposphere, further slowing the buildup. We modified the sentence to make this (hopefully) clearer:
"Events such as that in Beijing on day 20 are well simulated by the RFR model which is able to follow the full model, whereas the simulation without chemistry follows a distinctly different path that is solely determined by the net effects of emission, deposition, and (vertical and horizontal) transport."

*P9 L17: I would argue this is exactly the same of a chemical mechanism. There is no dependency between grid point especially as column depends (ozone) of the photolysis calculation has been taken out of the equation. Please clarify, why this is a unique feature of the ML approach.*
We refer here to the species independence within the same grid cell: each random forest (in fact, each single decision tree) can be evaluated independent from one another, whereas the ODEs of the chemical mechanism require coupling between species. We changed the wording to further clarify:
"A fundamental attractiveness of the random forest algorithm is its almost perfect parallel nature - even among species within the same grid cell: the nodes of all trees (and across all forests) solely depend on the initial values of the input features, and thus can be evaluated independent from one another (in contrast, the system of coupled ODEs solved by the chemical solver require coupling between the species)."

*P9 L20: Please comment on memory cost of the RFR implementation. The demands for floating point operation and memory determines the type of suitable architecture.*
The memory cost is highly dependent on the implementation of the RFR algorithm: since the same 1530 trees are evaluated in each grid point, the tree data need to be loaded only once as long as this information can be accessed efficiently by every CPU. For our test application, we loaded all information onto every CPU separately, which could be handled by our system.

*P10 L5: I think this point has a large potential, i.e. to select the complexity of the chemical mechanism according to location. In remote areas less demanding chemistry calculation might be sufficient, which are the majority of the grid points. (There might be issues with latency in the parallelism if specific points have more complex schemes to solve)*
Fully agreed. In fact, machine learning could be exploited to identify sub-mechanisms that minimize the number of mechanisms needed to capture the full variability of chemical regimes. We slightly modified the wording in section 4.4. to further clarify:

"The ability to represent the atmospheric chemistry as a set of individual machine learning models (one for each species) rather 20 than as one simultaneous integration has numerous advantages. In locations where the impact of a (relatively short-lived) molecule is known to be insignificant (for example isoprene over the polar regions or DMS over the deserts), the differential equation

approach continues to solve the chemistry for all species. However, with this machine learning methodology, there would be no need to call the machine learning algorithm for a species with a concentration below a certain threshold or for certain chemical environments (e.g. nighttime): the chemistry could continue without updating the change in the concentration 25 of these species. Thus it would be easy to implement a dynamical chemistry approach which uses a simple look-up table with predefined threshold rates to evaluate whether the concentration of a compound needs to be updated or not. If it did, the machine learning algorithm could be run, if it didn't the concentration would remain untouched and the evaluation of the random forest emulator is skipped (for this species). This approach could reduce the computational burden of atmospheric chemistry yet further."

*P10 L22: Please specify to what extent the current RFR and standard implementation conserves the stochimetry.*
The numeric implementation fulfills stoichiometry as it directly solves the system of chemical reactions. However, in our current implementation of the random forest there is no check for mass conservation / stoichiometric balance.

*P11 L4: Please provide more detail why this is easy and would be successful as small changes also accumulate over time.*
The proposed solution would simply skip the evaluation of the random forest for (short-lived) species in environments where the species is deemed insignificant, e.g. isoprene in remote areas. Typically, the concentrations in these locations are so small (and the life-time is so short) that ignoring the evaluation of isoprene under these circumstances has no impact on the overall chemistry. We updated the wording in this section to highlight this better, see comment above.

*P11 L22: Please quantify in more detail what amount of changes in emissions would be acceptable without need to retrain the RFR.*
Generally speaking, the RFR model should only be used under conditions that the model has seen during training. When changing the emissions, it should be checked that the maximum NOx/VOC ratio does not exceed the value seen during training. We added a clarifying statement to the revised version of the manuscript.

*P11 L26: Despite the computational cost there is a fundamental reason why ensemble approaches do not have the same importance in AQ forecast than in weather forecasting, namely the less chaotic dependence of AQ forecast on AQ initial conditions. But I agree that significantly cheap AC simulations would be of great benefit. For example, high resolution NWP model runs could afford to have AC modules embedded, which could be used to simulate the impact of aerosols and chemistry on the weather.*
AQ forecasts are very sensitive to the imposed boundary conditions, most notably emissions. Our hope is that ML-based ensemble forecasts can be used to capture a variety of emission scenarios (as long as they fall into the training range, as discussed above), as well as modifications to other chemical processes such as deposition or even photolysis. We added a sentence to the paragraph in the updated manuscript:

"Since air quality forecasts are much more sensitive to boundary conditions (e.g. emissions) than initial conditions, the different machine learning members would be used to capture the sensitivity of the air quality forecast to emission scenarios, changes in dry or wet deposition parameters, uncertainties in the chemical rate constants etc. Data-assimilation can be applied to determine the initial state for all models and then the ensembles could be used for probabilistic air quality

forecasting. This application is also less sensitive to long-term numerical instability of the machine learning model as the emulator is only used to produce 5-10 day forecasts, while the initial conditions are anchored to the full chemistry model for every new forecast."

---

## Editor Comment (EC1) · David Topping (Editor) · 19 Feb 2019

I would like to thank the authors and reviewers/readers of the presented study whilst briefly adding a rationale for why such papers fit within the journal's targeted audience. Studies on optimising the performance of, particularly, chemical mechanism solvers have been presented in various guises over many years. With the emergence of libraries that enabled rapid and agile prototyping of 'new' solutions that fit within the machine learning domain, it is important that the efficacy of a chosen workflow is clear. This is true whether the proposed solution demonstrates an improvement or reduction

in computational efficiency. Generating a discourse on this body of work is important and GMD now has clear provenance in publishing studies that demonstrate the advantages and disadvantages of alternative methods for model development. Under the ethos of the GMD publication, the code developed and evaluated in this study is available for others to use and I have no doubt this area will continue to grow and would encourage future studies that might offer improved solutions with regards to time-to-solution, using either the same family of algorithms or different methodologies and/or emerging hardware. With this in mind, the present study is now available for continuing this discourse and will no doubt generate much debate that leads to further submissions.